# The relationships of family income and caste-status with religiousness: Mediation role of intolerance of uncertainty

Sanjay Kumar[1]*, Martin Voracek[2]*

1 Department of Psychology, D.A.V. College, Muzaffarnagar, India, 2 Faculty of Psychology, Department of Cognition, Emotion, and Methods in Psychology, University of Vienna, Vienna, Austria

* sanjaykumar.paisal@gmail.com (SK); martin.voracek@univie.ac.at (MV)

## Abstract

The relationship between lower socioeconomic status (SES) and religiousness is well known; however, its (psychological mediation) mechanism is not clear. In the present study, we studied the mediation role of intolerance of uncertainty (IU; a personality measure of self-uncertainty) in the effect of SES on religiousness and its dimensions (i.e., believing, bonding, behaving, and belonging), in two different samples (students sample, $N = 868$, and community sample, $N = 250$), after controlling the effects of factors like age, sex, handedness, and self-reported risk-taking. The results showed that IU mediated the effects of lower family income and lower caste status (in students' sample only) on religiousness and its dimensions; higher caste status had a direct effect on religiousness (and its dimensions), and; among the sub-factors of IU, only prospective IU affected religiousness. Thus, along with showing that IU is a mediator of the effects of lower family income and lower caste status on religiousness, the present study supports the contention that religiousness is a latent variable that varied factors can independently initiate. Moreover, the present study suggests a nuanced model of the relationship between the hierarchical caste system and religiousness.

## Introduction

Religiousness is the relationship of an individual with transcendence. This relationship involves a combination of believing, bonding, behaving, and belonging [1]. Believing is a meaningful relationship, bonding is an emotional attachment, behaving is related to morality, and belonging is related to a self-esteem-enhancing affiliation [1]. Thus, encompassing cognitive, emotional, moral, and affiliation motives, religiousness provides a holistic framework (or life system), which, although not an actual representation of realities, is oriented towards coping with the realities of life (or living conditions; [2]). Such coping attempts may involve explanations of the deprivations as well as the compensations [3].

There is enormous support for the weaker version of the relationship between living conditions and religiousness (i.e., worse living conditions or socioeconomic deprivation or lower

**Competing interests:** The authors have declared that no competing interests exist.

socioeconomic status increases religiousness [4–9]), although some studies have supported the stronger version also (i.e., worse living conditions or socioeconomic deprivation produces religiousness [3,10]). Thus, perhaps the constraints and inadequacies (of a living condition) generate a psychological state that leads to (enhances or generates) religiousness. Lack of trust (i.e., psychological defensiveness; [11]), existential insecurity [5], self-uncertainty [6,8], and the lack of socio-psychological coping resources (i.e., distress; [12,13]) are some of the variables postulated to mediate the relationship between socioeconomic deprivation and religiousness. However, perhaps due to inadequacies of the conceptualization and measurement of the postulated mediation factors and the religiousness, the mechanics of the relationship between lower socioeconomic status (SES) and religiousness is unclear.

Studies have shown that, among individuals, the lower SES [9,11], difficult life conditions (living in sites of natural disasters, [10]; self-reported difficult living conditions; [14]), social exclusion [15], and lower income [8] are related to religiousness. Among countries also, underdevelopment [8], difficult life conditions (e.g., widespread hunger and low life expectancy; [14]) inequitable income distribution [4,6,7], and low economic development [4,8] are related to religiousness. Thus, socioeconomic deprivation (which normally reflects the worse living conditions) affects religiousness among both the individuals and the nations; however, the effect of socioeconomic factors on religiousness is much stronger at the level of nations (among countries), than at the level of individuals (among individuals; [5,14]). Therefore, perhaps some psychological factors that show differences between developed and developing countries are important for understanding the SES-religiousness connection. Recently, studies have focused on such psychological variables (i.e., varying with the socioeconomic differences between countries; e.g., psychological defensiveness [11]; uncertainty stress [16]).

Compared to developed countries, psychological defensiveness is higher in developing countries [11,17]. Therefore, it may be a psychological reflector of the between-countries socioeconomic differences. However, whereas studies have reported that psychological defensiveness (opposite of trust) mediates the effect of lower SES on religiousness [11], this effect is salient in developed countries only [11]. Thus, because developing countries have more socioeconomic disparity and more absolute deprivation (compared to developed countries), psychological defensiveness only partially explains the effect of socioeconomic factors on religiousness.

Studies have suggested that existential insecurity (i.e., vulnerabilities to societal and personal risks and threats) mediates the effect of difficult living conditions on religiousness among nations [5]. However, existential security has been inferred only through self-reported, or the available, socioeconomic facilities [5]. Moreover, whereas low life expectancy supports increased fear of death among low SES individuals [18], studies have rendered, at best, only weak support for the relationship between fear of death and religiousness [19].

Conceptually, lack of trust (i.e., psychological defensiveness; [20]) and existential insecurity are manifestations of self-uncertainty (i.e., people being uncertain about themselves; [21]). Thus, the latter is likely to be the more basic psychological factor in the relationship of lower SES with religiousness. Self-uncertainty is related to the hardening of religious attitudes [22,23], and is suggested to be the basis for identification with religious groups (uncertainty-identity theory; [24]). Studies have suggested that self-uncertainty is a mediator in the effects of lower income on religiousness [6,8]. However, as far as we know, this likelihood has not been directly tested, except for a study on the relationship of social exclusion (a condition presumably related to lower SES) with religiousness [15].

In prior studies, self-uncertainty either was generated experimentally [22,23], was measured through partially standardized questionnaires [15], or was presumed (to be occurring) through related conditions (i.e., low economic development, income insecurity, and health insecurity

[4]). Experimentally generated uncertainties tend to reflect transient states and have limited ecological validity [25], whereas socioeconomic factors are long-term variables with lasting effects [26]. Thus, the study of personality measure of self-uncertainty (as a mediator of the effect of socioeconomic factors on religiousness) may be more fruitful.

Moreover, studies have shown that, compared to higher status individuals, lower SES individuals show more psychological distress (irrespective of the similarity of the exposed stressful situation; [13]), which, in turn, is related to religiousness [27]. However, because psychological distress is a symptom (i.e., unspecified stress, anxiety, and depression), it has limited explanatory power (for understanding the underlying psychological processes). Moreover, studies have shown that intolerance of uncertainty underlies psychological distress [28,29].

Intolerance of uncertainty (IU) is "an individual's dispositional incapacity to endure the aversive response triggered by the perceived absence of salient, key, or sufficient information, and sustained by the associated perception of uncertainty"(p31) [30]. Thus, rather than simply reflecting the volume of uncertainty (a person is facing), as a personality measure, IU captures the general inability to face the uncertainty. Compared to other related concepts, IU is a better measure of uncertainty [31]. It measures fear of uncertainty, which is conceptually the basic fear (pervading most other fears and threats like fear of death; [32]). Compared to developed countries, the developing countries (with lower GDP) have much higher IU (i.e., mean IUS-12 score is higher by approximately a standard deviation in Brazil [33], China [34], and India [35]). IU is a trait measure (and not a type measure) that may be used to study individual differences (in the general population; [36]). It is measurable through highly reliable and cross-culturally valid scales [33–35,37]. Moreover, IU is related to religiousness [38,39] and is, conceptually, a personality measure of self-uncertainty [30]. Thus, as discussed above, IU underlies several proposed mediation factors (e.g., psychological distress, existential insecurity, fear of death, psychological defensiveness, and self-uncertainty) of the relationship between lower SES and religiousness and has the potential to reflect the socioeconomic differences between countries also. Moreover, because increased future uncertainties are characteristic of lower SES [40]; the SES translates into personality [41]; IU is a dispositional reaction to future uncertainties [30], and; religiousness is an attempt to cope with the future uncertainties [42], theoretically also, the lower SES, IU, and religiousness are interrelated through the common denominator of future uncertainties. Therefore, IU may be an effective mediator of the relationship between lower SES and religiousness.

Furthermore, in most of the prior studies (i.e., on the relationship between lower SES and religiousness), religiousness was measured either through behavioral indicators [6,7], beliefs [4], or a combination of them [5,8,9,14]. Although some earlier studies do recognize the multidimensional nature of religiousness [43], the recent theoretical advances support a composite four-dimensional conception of religiousness (i.e., believing, bonding, behaving, and belonging), which is likely to be universal and is cross-culturally and cross-religiously valid [44,45]. This conception seeks to discriminate religiousness from other related concepts, like non-organized spirituality, by emphasizing that only religiousness encompasses all the four dimensions, i.e., believing, bonding, behaving, and belonging [1,45]. Moreover, the dimensions (of religiousness) are value neutral and may study functional vs. dysfunctional aspects of religion through the qualitative analysis [1].

Thus, the present study perhaps is an initial study of the effect of SES on the four-basic dimensional conceptualization of religiousness. Because religiousness encompasses different dimensions, it would be interesting to study whether the effect of SES (and other variables) on general religiousness translates into its different dimensions also. In case the same effect expresses in all the four dimensions of religiousness, according to this conception (i.e., religiousness is a composite of four dimensions), the confidence in the occurrence of this effect on

religiousness will be stronger (i.e., this effect is not occurring on the other related concepts like spirituality). Moreover, because, along with quantitative analysis, this (multidimensional) conception of religiousness allows for qualitative analyses (inter-correlations among dimensions and salience of dimensions in a group; [45]), in the present study, we have conducted such analysis also (discussed below).

Generally, the SES involves variables like income, social status, education, and access to health facilities. In India, caste status is an important SES variable [46]. It is an ascribed status, which represents a composite hierarchy of ritual, social, and economic scales [46,47]. It is determined by birth in an endogamous group [46]. The caste was originally (historically) characteristic of Hindus but presently it is characteristic of other religious groups as well [48,49]. In modern India, castes are functionally grouped in three descending hierarchical categories: upper, higher, or general castes (GC), intermediate or other backward castes (OBC), and scheduled castes (SC; [47]) and, irrespective of some transformations (i.e., weakening of *Jajmani* system, the caste-occupation connection), caste is quite effectively a psycho-social reality (i.e., endogamous, hierarchical, and psychologically felt; [50–52]).

As far as we know, irrespective of a large number of observational and anecdotal studies on the relationship of caste with religion, there are few studies on the quantitative relationship of caste status with religiousness. The caste hierarchy is based on the ideology of purity and pollution, which is supported by Hindu scriptures (classifying caste groupings based on occupations; [46]). Thus, isolation and discrimination are integral to the caste system of social organization [53]. Studies have shown that personal uncertainty mediates the effect of isolation and discrimination on religiousness [15]. Therefore, the mediation effect of IU on the relationship of lower caste status (which generates a feeling of isolation and discrimination) with religiousness is expected. Because India is a modern market-based economy, family income may be an important component of SES. Moreover, education (another indicator of SES) is also related to lower religiousness [7,54]. Although the relationship of education with IU is unknown, because education involves exposure to multiple viewpoints, we expect that education has a decremental effect on IU, which, in turn, leads to lower religiousness. Thus, in the present study, we studied the mediation role of IU on the effects of caste groups, family income, and education (as components of SES) on religiousness.

Studies have reported that self-uncertainty mediates the effect of minority communities on religiousness [15]. Moreover, different religions may have qualitatively different religiousness [45]. Similarly, age (religiousness [55]; IU [56]), women (religiousness [57]; IU [58]), self-reported risk-taking (religiousness [59]; lower IU [58]), and right-handedness (religiousness [60]; IU [61]) are related to both religiousness and IU. Thus, these factors may be confounding the mediation role of IU in the effect of SES on religiousness. Studies have reported that height [59] and body mass index [62] are also factors in religiousness.

Because IU [63] and religiousness [64,65] are related to prefrontal cortex; prefrontal cortex structures are amenable to developmental processes [66]; SES [67], risk-taking [68], sex and age [69] are related to prefrontal cortex and other cerebral cortex areas, and; handedness [70] and height [71] are related to cortical surface areas, the seemingly different antecedent factors (i.e., the bio-socio-developmental factors, e.g., self-reported risk-taking, age, height, handedness etc. vs. the pure environmental factors, e.g., SES) may be affecting the IU-religiousness connection through effects on the brain structure. Thus, although, in the present study, we are not studying the effects on the brain, this framework of understanding (i.e., mediation role of the brain structure) does justify conducting a comprehensive study of the effects of different antecedent variables (environmentally-determined as well as bio-socio-developmental factors) on IU-religiousness connection. Such analyses may generate new understandings for future research.

In the present study, we studied two different samples, i.e., a students' sample and a community sample. Initially, the hypotheses were established in a larger sample of students and, subsequently, replication tests were done in the community sample. This is an ecologically valid testing strategy that renders robust effects [72]. To be precise, the following hypotheses were tested in the present study: (1) IU mediates the effect of lower family income on religiousness; (2) IU mediates the effect of lower caste status on religiousness, and; (3) IU mediates the effect of lower education on religiousness. Moreover, because earlier studies have shown that, compared to western and monotheistic religions (r = .70 to .84), eastern and non-monotheistic religions have low inter-correlations (r = .31 to .68) among the dimensions of religiousness [45], we hypothesized that (4) in the Indian population (with the predominance of non-monotheistic Hindu religion), the inter-correlations in the dimensions of religiousness are weaker (below .70). Moreover, in the present study, we assessed the interrelationships (in the dimensions of religiousness) as well as the relative importance of the dimensions of religiousness in different religions, castes, and income groups, albeit without any directional hypothesis. Thus, the present study is a comprehensive report on the IU-mediated effects (as well as the direct effects) of SES and other variables on religiousness and its dimensions, which includes qualitative analyses also.

## Method

The study (part of a project: IU and its correlates) was conducted in academic session 2017–18 and was approved by the local ethics committee of the institution (i.e., D.A.V. College Ethics Committee).

### Participants

**Students' sample.** A sample of 868 students was selected from a college in Muzaffarnagar city of western Uttar Pradesh, India. The participants were selected based on availability as well as were actively approached, sometimes with the help of faculty members.

**Community sample.** A sample of 250 participants was selected from Muzaffarnagar city. Research assistants approached participants through personal networks and referrals. Higher age participants were preferred and the cutoff age was 25 years.

### Material and procedure

Before the assessment, participants gave written informed consent. The research assistant gave detailed instructions and participants filled a set of paper-pencil formatted questionnaires individually (mostly in community sample) as well as in small groups (less than 40 participants).

In the personal details section, participants informed age, height, weight, sex, education level (below intermediate, undergraduate, postgraduate, higher degree), monthly family income (in rupees; less than 10 thousand, 10–30 thousand, 30–60 thousand, 60 thousand-1 Lakh, above 1 lakh), and caste category (SC, OBC, GC). For the disbursement of benefits of various schemes, the admissions to educational institutes, and employment, people are frequently required to fill their caste categories in India [47]. Thus, asking participants their caste category is an established common procedure for determining the caste categories [47,73].

For the assessment of IU, a reliable (Cronbach's $\alpha$ = .79) and valid Hindi translation version of the IU scale small form (IUS-12) was used [35]. The IUS-12 is a 12-item scale with a five-point response scheme (1 = *strongly disagree* to 5 = *strongly agree*) so that a higher score reflects more IU [37]. It measures total IU as well as the components: prospective IU and inhibitive IU. Although a correlated two-factor structure of IU (prospective IU and inhibitive IU; [74])

has been reported, the bi-factor model studies strongly support studying IU as a unitary concept (the general factor of IU; [75]).

To assess religiousness, a reliable (α = .88) and valid Hindi translation version of the 4BDRS was used (a 12-item scale; [44]). The 4BDRS is a test of general religiousness with cross-cultural and cross-religious validity [44,45]. It has a seven-point response scale (1 = *strongly disagree* to 7 = *strongly agree*) attached with each item so that a higher score indicates more religiousness. The 4BDRS measures the general religiousness as well as the four basic dimensions of religiousness: believing, bonding, behaving, and belonging, each of which is measured through three separate items [45]. Moreover, as suggested by Saroglou et al. [45], we transformed the raw scores of general religiousness and dimensions of religiousness into single item scores (total score/ number of items).

For assessment of handedness, we used a 14-item hand preference questionnaire, procedural details and reliability of which are reported elsewhere [61]. It has a five-point response scale (1 = *always left hand* to 5 = *always right hand*) attached with each item. The laterality quotient (LQ), ranging from -100 to 100, reported the degree of handedness when the negative values indicate left-handedness and the positive values indicate a right-handedness.

The assessment of risk-taking propensity was done through a single item similar to that used for establishing the genetics of self-reported risk-taking [76]. The item reads "I enjoy risk-taking?" [59]. A 7-point response scale (1 = *fully incorrect about me* to 7 = *fully correct about me*) was attached. Thus, a higher score indicates more risk-taking.

## Statistical procedure

Mediation path analysis was conducted through Process procedure, in IBM SPSS 20, with 5000 bootstrap cycles for estimation of 95% confidence interval and standard error. First, the model with the total or general IU as a single mediation variable and the general religiousness as a dependent variable, second, the model with the prospective IU and inhibitive IU as two mediation variables and the general religiousness as a dependent variable, and lastly, the model with the general IU as a single mediation variable and the dimension of religiousness as a dependent variable (separate analysis for each dimension) were conducted. The interrelationship in the dimensions of religiousness (i.e., believing, bonding, behaving, and belonging), among groups, were studied through simple and partial correlations. The repeated measure analysis was used to assess the relative importance attributed to each dimension of religiousness, among groups. The same steps were followed for the statistical analysis of both samples.

## Results

### Students' sample

Table 1 shows the distribution of variables. After the list-wise deletion of missing cases, path analyses were conducted on the sample of 838 cases.

**General IU as the mediator of the effects of antecedent variables on general religiousness.** In the first model, general IU was the mediation factor in the relationship of SES (family income and caste status) and other variables (covariates) with religiousness. Fig 1 shows the final path analysis model. Table 2 shows the effects of antecedent variables on the mediation variable as well as (direct and indirect effect) on the dependent variable.

Although the omnibus direct effect of income on religiousness was (nominally) non-significant, $F(4, 819) = 2.3$, $p = .057$, participants in the family income bracket of 60 thousand to 1lakh does show lower religiousness than participants with income lower than this (lower than 60 thousand; relative direct effect). Moreover, there was a significant (IU mediated) indirect effect of family income (30–60 thousand vs. lower income) on religiousness. Thus, the decrease

**Table 1. The distribution of variables in the students sample.**

| Variable | M | SD | Range | N |
|---|---|---|---|---|
| Religiousness | 5.20 | 1.53 | 1–7 | 861 |
| Believing | 5.20 | 1.81 | 1–7 | 864 |
| Bonding | 5.44 | 1.79 | 1–7 | 866 |
| Behaving | 5.04 | 1.89 | 1–7 | 863 |
| Belonging | 5.10 | 1.99 | 1–7 | 866 |
| IU | 40.64 | 9.70 | 15–60 | 862 |
| Prospective IU | 25.23 | 5.78 | 9–35 | 867 |
| Inhibitive IU | 15.41 | 4.98 | 5–25 | 863 |
| Age | 21.43 | 3.12 | 16–29 | 868 |
| Sex (female) | 0.48 | 0.50 | 0–1 | 868 |
| Handedness (LQ) | 77.65 | 38.38 | -100-100 | 867 |
| Height (in meters) | 1.64 | 0.10 | 1.07–1.98 | 861 |
| BMI | 32.56 | 5.43 | 18.37–52.28 | 857 |
| Caste status (OBC) | 0.45 | 0.50 | 0–1 | 861 |
| Caste status (SC) | 0.27 | 0.45 | 0–1 | 861 |
| Income (60 thousand-1lakh) | 0.09 | 0.29 | 0–1 | 865 |
| Income (30–60 thousand) | 0.22 | 0.42 | 0–1 | 865 |
| Income (10–30 thousand) | 0.42 | 0.49 | 0–1 | 865 |
| Income (< 10 thousand) | 0.23 | 0.42 | 0–1 | 865 |
| Religion (muslim) | 0.12 | 0.32 | 0–1 | 868 |
| Religion (others) | 0.07 | 0.26 | 0–1 | 868 |
| Risktaking score (2) | 0.04 | 0.20 | 0–1 | 859 |
| Risktaking score (3) | 0.06 | 0.24 | 0–1 | 859 |
| Risktaking score (4) | 0.17 | 0.38 | 0–1 | 859 |
| Risktaking score (5) | 0.07 | 0.26 | 0–1 | 859 |
| Risktaking score (6) | 0.08 | 0.26 | 0–1 | 859 |
| Risktaking score (7) | 0.38 | 0.49 | 0–1 | 859 |
| Education (PG & above)* | 0.25 | 0.44 | 0–1 | 865 |

*Note.* IU = intolerance of uncertainty, LQ = laterality quotient, BMI = body mass index, OBC = other backward castes, SC = scheduled castes, PG = post-graduation. *N* shows the number of participants after deletion of missing cases.

* Because of fewer cases, extreme categories merged to form two categories' of 'Undergraduate & below' and 'PG & above'.

in family income leads to an increase in IU, which, in turn, leads to an increase in religiousness.

Caste had both direct and indirect effects. Thus, lowering of caste status (GC vs. others) leads to an increase in IU, which, in turn, leads to an increase in religiousness (indirect effect). However, OBC showed more religiousness than SC (direct effect; omnibus test: $F(2, 819) =$ 5.95, $p = .003$).

Age had an indirect effect, but no direct effect, on religiousness. Thus, an increase in age leads to higher IU, which, in turn, leads to higher religiousness. Similar to this, handedness also had an indirect effect, but no direct effect, on religiousness. Thus, right-handedness leads to higher IU, which, in turn, leads to higher religiousness.

Risk-taking had a direct (omnibus test: $F(6, 819) = 3.79$, $p = .001$) as well as an indirect effect on religiousness. Thus, highest-level risk-takers (scoring 7, compared to lowest-level risk-

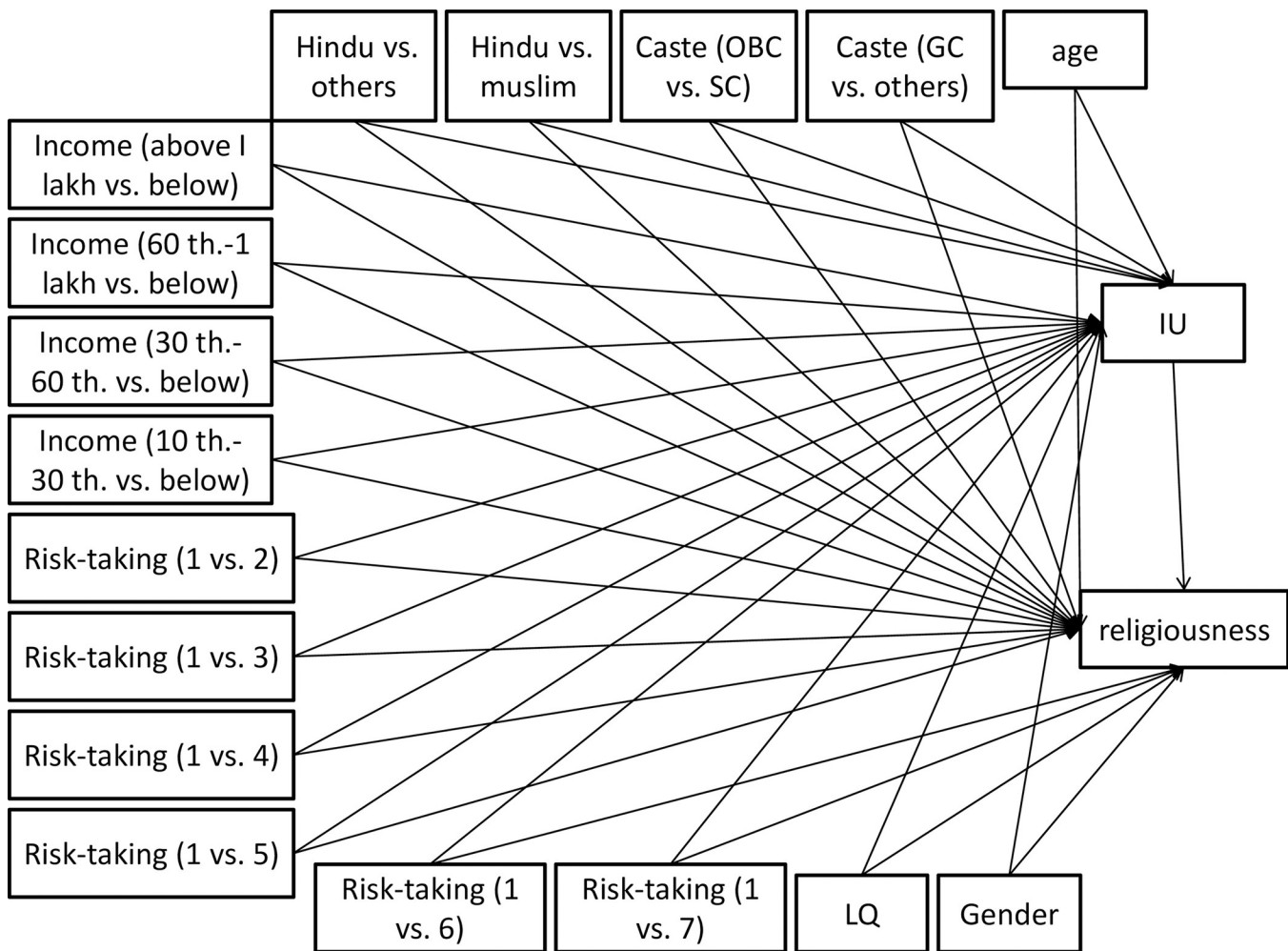

**Fig 1. The conceptual path diagram shows the studied relationships of antecedent variables with IU and religiousness in the students' sample.**
IU = intolerance of uncertainty, GC = general castes, OBC = other backward castes, SC = scheduled castes, th. = thousand, LQ = laterality quotient (handedness).

takers) have higher religiousness (direct effect), whereas intermediate-level risk-takers (scoring 5, compared to lowest-level risk-takers) have lower IU, which, in turn, leads to lower religiousness (indirect effect).

Sex had a direct, but no indirect, effect on religiousness. Thus, women (compared to men) have higher religiousness. Similarly, religion had a direct (omnibus test: $F(2, 819) = 15.95$, $p < .001$), but no indirect, effect on religiousness. Thus, minority-religion participants (compared to majority-religion participants) have higher religiousness. There was no moderation effect (X-M interaction) in any of the above-reported mediation analyses (Table 2).

Because height is collinear with the combination of sex, age, and caste, the effect of height was separately analyzed for men and women (see S1 Fig). An indirect effect ($b = -.75$, SE = 0.3, 95% CI = -1.5 to -0.2; direct effect: $b = 0.46$, $t(375) = 0.5$, $p = .6$; X-M interaction: $F(1, 374) = 0.3$, $p = .6$) of height on religiousness among women (but not among men, indirect effect: $b = 0.10$, SE = 0.26, 95% CI = -0.4 to 0.6; direct effect: $b = 1.1$, $t(419) = 1.2$, $p = .6$; X-M interaction: $F(1, 418) = 1.1$, $p = .3$) was found. Thus, taller women have lower IU which, in turn, leads to lower religiousness.

**Table 2. The effects (direct and indirect effects) of antecedent variables on the mediation variable (general IU) and the dependent variable (religiousness) in the students sample.**

| Antecedent variable | Effect on M | | | Direct effect on Y | | | Indirect effect on Y | | | |
|---|---|---|---|---|---|---|---|---|---|---|
| | Coeff. | SE | t | Coeff. | SE | t | Coeff. | SE | LCI | HCI |
| Intolerance of uncertainty | | | | 0.52 | 0.06 | 8.3*** | | | | |
| Caste (GC vs. others) | 1.69 | 0.76 | 2.2* | -0.86 | 1.36 | -0.6 | **0.88** | 0.41 | 0.11 | 1.72 |
| Caste (OBC vs.SC) | -0.23 | 0.83 | -0.3 | -5.13 | 1.49 | -3.5*** | -0.12 | 0.45 | -1.02 | 0.74 |
| X*M (X = caste) | | | | | | | $F(2, 817) = 0.8, p = .5$ | | | |
| Income (above vs. below 1lakh) | 1.035 | 1.69 | 0.6 | 4.59 | 3.01 | 1.5 | 0.54 | 0.93 | -1.25 | 2.38 |
| Income (60 thousand-1Lakh vs. below) | 1.97 | 1.17 | 1.7 | 4.31 | 2.09 | 2.1 * | 1.02 | 0.63 | -0.15 | 2.36 |
| Income (30–60 thousand vs. below) | 1.78 | 0.83 | 2.1* | -0.19 | 1.48 | -0.1 | **0.92** | 0.45 | 0.07 | 1.82 |
| Income (10–30 thousand vs. below) | 0.65 | 0.89 | 0.7 | -1.74 | 1.58 | -1.1 | 0.34 | 0.48 | -0.59 | 1.29 |
| X*M (X = income) | | | | | | | $F(4, 815) = 0.7, p = .6$ | | | |
| Handedness (LQ) | 0.028 | 0.01 | 3.2 *** | 0.003 | 0.015 | 0.2 | **0.014** | 0.01 | 0.01 | 0.03 |
| X*M (X = Handedness) | | | | | | | $F(1, 818) = 0.3, p = .6$ | | | |
| Sex | 0.89 | 0.68 | 1.3 | 4.23 | 1.2 | 3.5 *** | 0.46 | 0.36 | -0.22 | 1.18 |
| X*M (X = Sex) | | | | | | | $F(1, 818) = 1.0, p = .3$ | | | |
| Risk-taking (1vs 2) | -1.33 | 1.86 | -0.7 | -1.13 | 3.3 | -0.3 | -0.69 | 1.12 | -2.98 | 1.42 |
| Risk-taking (1vs 3) | -2.64 | 1.55 | -1.7 | 1.65 | 2.76 | 0.6 | -1.37 | 0.84 | -3.05 | 0.26 |
| Risk-taking (1vs 4) | -0.15 | 1.1 | -0.1 | 2.07 | 1.97 | 1.1 | -0.08 | 0.58 | -1.22 | 1.07 |
| Risk-taking (1vs 5) | -4.69 | 1.44 | -3.3*** | -0.37 | 2.58 | -0.1 | **-2.44** | 0.78 | -4.05 | -1.0 |
| Risk-taking (1vs 6) | -0.49 | 1.43 | -0.3 | -1.43 | 2.55 | -0.6 | -0.25 | 0.78 | -1.8 | 1.29 |
| Risk-taking (1vs 7) | -0.11 | 0.94 | -0.1 | 5.91 | 1.67 | 3.5 *** | -0.06 | 0.50 | -1.04 | 0.94 |
| X*M (X = Risk-taking) | | | | | | | $F(6, 813) = 1.2, p = .3$ | | | |
| Age | 0.38 | 0.11 | 3.5 *** | 0.33 | 0.20 | 1.7 | **0.20** | 0.07 | 0.072 | 0.34 |
| X*M (X = Age) | | | | | | | $F(1, 818) = 0.2, p = .6$ | | | |
| Religion (Hindu vs. muslim) | -1.04 | 1.06 | -1.0 | 5.78 | 1.89 | 3.1 ** | -0.54 | 0.51 | -1.62 | 0.43 |
| Religion (Hindu vs. other religion) | 0.95 | 1.31 | 0.7 | 11.46 | 2.32 | 4.9 *** | 0.49 | 0.74 | -0.91 | 2.02 |
| X*M (X = Religion) | | | | | | | $F(2, 817) = 1.1, p = .3$ | | | |
| | $R^2 = 0.071, F(17, 820) = 3.66, p < .001$ | | | $R^2 = 0.18, F(18, 819) = 9.83, p < .001$ | | | | | | |

Note. M = mediation variable (intolerance of uncertainty), Y = Dependent variable (religiousness), Coeff. = coefficient, SE = standard error, LCI = lower limit of 95% confidence interval, UCI = upper limit of 95% confidence interval, GC = general castes, OBC = other backward castes, SC = scheduled castes, LQ = laterality quotient, X*M = interaction of antecedent and mediation variables (moderation effect); bold values indicate significant indirect effects (not including zero between upper and lower limits of 95% confidence interval).

*** Significant at the .001 level.

** Significant at the .01 level.

* Significant at the .05 level.

Moreover, because BMI (indirect effect: $b = 0.005$, SE = 0.003, 95% CI = -0.001 to 0.01; direct effect: $b = -0.01$, $t(825) = -1.4$, $p = .2$) and education (indirect effect: $b = 0.04$, SE = 0.04, 95% CI = -0.3 to 0.11; direct effect: $b = 0.21$, $t(835) = 1.8$, $p = .07$) had no effect on religiousness, these variables were excluded from the final model.

**Prospective and inhibitive IUs as mediators of the effects of antecedent variables on general religiousness.** There were no mediation effects of the combination of prospective and inhibitive IUs. Therefore, we have reported the results of a simpler, parallel mediation, model (analyzing mediation effects of prospective and inhibitive IUs; see Table 3 and S2 Fig).

The lower family income (30–60 thousand vs. lower income) was related to inhibitive IU; however, it failed to translate into an (indirect) effect on religiousness. The lower caste status

**Table 3. The effects (direct and indirect) of antecedent and mediation (prospective IU & inhibitive IU) variables on religiousness in the students' sample.**

| | Effect on M1 | | | Effect on M2 | | | Direct effect on Y | | | Indirect effect (on Y) mediated by M1 | | | | Indirect effect (on Y) mediated by M2 | | | |
|---|---|---|---|---|---|---|---|---|---|---|---|---|---|---|---|---|---|
| | Cof. | SE | $t$ | Cof. | SE | $t$ | Cof. | SE | $t$ | Cof. | SE | LCI | UCI | Cof. | SE | LCI | UCI |
| IUP | | | | | | | 0.9 | 0.13 | 6.8*** | | | | | | | | |
| IUI | | | | | | | 0.07 | 0.15 | 0.5 | | | | | | | | |
| C1 | 0.96 | 0.45 | 2.1* | 0.73 | 0.39 | 1.9 | -0.89 | 1.35 | -0.7 | **0.86** | 0.42 | 0.07 | 1.71 | 0.05 | 0.13 | -0.2 | 0.36 |
| C2 | 0.13 | 0.50 | 0.3 | -0.36 | 0.43 | -0.8 | -5.34 | 1.48 | -3.6*** | 0.12 | 0.47 | -0.79 | 1.06 | -0.03 | 0.09 | -0.3 | 0.14 |
| X*M | | | | | | | | | | $F(2, 816) = 0.2, p = .8$ | | | | $F(2, 816) = 0.8, p = .4$ | | | |
| I1 | 0.69 | 1.00 | 0.7 | 0.34 | 0.88 | 0.4 | 4.48 | 2.99 | 1.5 | 0.62 | 1.03 | -1.37 | 2.72 | 0.02 | 0.15 | -0.25 | 0.39 |
| I2 | 1.15 | 0.70 | 1.7 | .82 | .61 | 1.4 | 4.24 | 2.07 | 2.1* | 1.03 | 0.68 | -0.21 | 2.48 | 0.06 | 0.16 | -0.27 | 0.42 |
| I3 | 0.86 | 0.49 | 1.8 | 0.91 | 0.43 | 2.1* | -0.1 | 1.47 | -0.1 | 0.77 | 0.47 | -0.14 | 1.73 | 0.07 | 0.16 | -0.23 | 0.43 |
| I4 | 0.40 | 0.53 | 0.8 | 0.25 | 0.46 | 0.6 | -1.78 | 1.57 | -1.1 | 0.36 | 0.47 | -0.52 | 1.32 | 0.02 | 0.08 | -0.14 | 0.22 |
| X*M | | | | | | | | | | $F(4, 814) = 0.7, p = .6$ | | | | $F(4, 814) = 0.7, p = .6$ | | | |
| LQ | 0.012 | 0.005 | 3.8*** | 0.008 | 0.004 | 1.9 | -0.0 | 0.015 | -0.0 | **0.017** | 0.006 | 0.01 | .03 | 0.001 | 0.002 | -0.002 | 0.004 |
| X*M | | | | | | | | | | $F(1, 817) = 0.2, p = .6$ | | | | $F(1, 817) = 0.7, p = .4$ | | | |
| Age | 0.26 | 0.06 | 4.0*** | 0.12 | 0.06 | 2.1* | 0.28 | 0.19 | 1.5 | **0.23** | 0.07 | 0.11 | 0.38 | 0.01 | 0.02 | -0.03 | 0.05 |
| X*M | | | | | | | | | | $F(1, 817) = 0.2, p = .7$ | | | | $F(1, 817) = 0.1, p = .8$ | | | |
| RT1 | -1.04 | 1.1 | -0.9 | -0.29 | 0.96 | -0.3 | -0.86 | 3.29 | -0.3 | -0.93 | 1.19 | -3.5 | 1.24 | -0.02 | 0.17 | -0.42 | 0.29 |
| RT2 | -1.62 | 0.92 | -1.8 | -1.02 | 0.80 | -1.3 | 1.80 | 2.74 | 0.7 | -1.45 | 0.88 | -3.31 | 0.23 | -0.07 | 0.20 | -0.56 | 0.31 |
| RT3 | -0.16 | 0.66 | -0.2 | 0.01 | 0.57 | 0.0 | 2.14 | 1.95 | 1.1 | -0.14 | 0.60 | -1.35 | 1.04 | 0.00 | 0.09 | -0.20 | 0.22 |
| RT4 | -2.3 | 0.86 | -2.7** | -2.4 | 0.75 | -3.2** | -0.57 | 2.57 | -0.2 | **-2.06** | 0.81 | -3.69 | -0.51 | -0.18 | 0.38 | -0.98 | 0.56 |
| RT5 | 0.61 | 0.85 | 0.7 | -1.1 | 0.74 | -1.5 | -2.2 | 2.54 | -0.9 | 0.55 | 0.81 | -1.09 | 2.11 | -0.08 | 0.21 | -0.58 | 0.29 |
| RT6 | 0.31 | 0.56 | 0.6 | -0.42 | 0.49 | -0.9 | 5.60 | 1.67 | 3.4*** | 0.28 | 0.52 | -0.75 | 1.31 | -0.03 | 0.10 | -0.29 | 0.16 |
| X*M | | | | | | | | | | $F(6, 812) = 1.3, p = .3$ | | | | $F(6, 812) = 0.7, p = .7$ | | | |
| R1 | -0.20 | 0.63 | -0.32 | -0.84 | 0.55 | -1.5 | 5.49 | 1.88 | 2.9** | -0.18 | 0.55 | -1.32 | 0.88 | -0.61 | 0.15 | -0.43 | 0.22 |
| R2 | -0.14 | 0.78 | -0.18 | 1.09 | 0.68 | 1.61 | 12.0 | 2.32 | 5.2*** | -0.13 | 0.75 | -1.60 | 1.34 | 0.08 | 0.21 | -0.30 | 0.58 |
| X*M | | | | | | | | | | $F(2, 816) = 0.29, p = .8$ | | | | $F(2, 816) = 2.0, p = .13$ | | | |
| Sex | 0.64 | 0.40 | 1.6 | 0.25 | 0.35 | 0.7 | 4.1 | 1.20 | 3.4*** | 0.58 | 0.37 | -0.12 | 1.34 | 0.02 | 0.07 | -0.11 | 0.19 |
| X*M | | | | | | | | | | $F(1, 817) = 0.0, p = .99$ | | | | $F(1, 817) = 3.34, p = .07$ | | | |
| | $R^2 = 0.08$, $F(17, 820) = 3.91$, $p < .001$ | | | $R^2 = 0.05$, $F(17, 820) = 2.57$, $p < .001$ | | | $R^2 = 0.19$, $F(19, 818) = 9.98$, $p < .001$ | | | | | | | | | | |

*Note*. M1 = mediation variable 1 (IUP), M2 = mediation variable 2 (IUI), Y = Dependent variable (religiousness), Cof. = coefficient, SE = standard error, LCI = lower limit of 95% confidence interval, UCI = upper limit of 95% confidence interval, IUP = prospective intolerance of uncertainty, IUI = inhibitive intolerance of uncertainty, C1 = general castes vs. others, C2 = other backward castes vs. scheduled castes, I1 = Income (above vs. below 1lakh), I2 = Income (60 thousand-1Lakh vs. below), I3 = Income (30–60 thousand vs. below), I4 = Income (10–30 thousand vs. below), LQ = laterality quotient, RT1 = Risk-taking (1vs. 2), RT2 = Risk-taking (1vs. 3), RT3 = Risk-taking (1vs. 4), RT4 = Risk-taking (1vs. 5), RT5 = Risk-taking (1vs. 6), RT6 = Risk-taking (1vs. 7), R1 = Hindu vs. Muslim, R2 = Hindu vs. others, X*M = interaction of antecedent (given in above row) and mediation variables (moderation effect); bold values indicate significant indirect effects (not including zero between upper and lower limits of 95% confidence interval).

*** Significant at the .001 level.

** Significant at the .01 level.

* Significant at the .05 level.

(GC vs. others) and the right-handedness were related to prospective IU and these effects translated into (indirect) effects on religiousness. Age and the intermediate level of risk-taking (scoring 5, compared to lowest-level risk-takers) were related to both prospective and inhibitive IUs, however, only the effects on prospective IU translated into (indirect) effects on religiousness. Thus, between the sub-factors of IU, only prospective IU was a factor in the religiousness (and not the inhibitive IU). The direct effects of antecedent variables (i.e., the

direct effects of higher caste status, lower family income, the highest level of risk-taking, minority religions, and women) on religiousness were similar to that reported in the above section.

Moreover, among women, whereas height affected both inhibitive IU ($b$ = -8.9, $t(376)$ = -2.45, $p$ = .02) and prospective IU ($b$ = -11.8, $t(376)$ = -2.9, $p$ = .004), only the effect on prospective IU translated into an effect on religiousness (indirect effect: $b$ = -1.09, SE = 0.45, 95% CI = -2.1 to -0.3).

**General IU as the mediator of the effects of antecedent variables on the dimensions of religiousness.** We also studied the effects of antecedent variables and general IU (mediation factor) on the dimensions of religiousness (for statistics, see Tables 4–7). The effects were largely similar to the effects reported for the general religiousness above.

**Table 4. The effects (direct and indirect effect) of antecedent and mediation (general IU) variables on believing in the students' sample.**

| Antecedent variable | Effect on M | | | Direct effect on Y | | | Indirect effect on Y | | | |
|---|---|---|---|---|---|---|---|---|---|---|
| | Coeff. | SE | $t$ | Coeff. | SE | $t$ | Coeff. | SE | LCI | UCI |
| Intolerance of uncertainty | | | | 0.14 | 0.02 | 7.6*** | | | | |
| Caste (GC vs. others) | 1.69 | 0.76 | 2.23* | 0.19 | 0.41 | 0.46 | **0.24** | .1117 | .0259 | .4702 |
| Caste (OBC vs.SC) | -0.23 | 0.84 | -0.28 | -1.13 | 0.45 | -2.52* | -.033 | .1258 | -.2951 | .2080 |
| X*M (X = caste) | | | | | | | $F(2, 817)$ = 1.04, $p$ = .4 | | | |
| Income (above vs. below 1lakh) | 1.04 | 1.69 | 0.6 | 2.2 | 0.91 | 2.5* | 0.15 | 0.25 | -0.34 | 0.65 |
| Income (60 thousand-1Lakh vs. below) | 1.97 | 1.17 | 1.69 | 0.75 | 0.63 | 1.2 | 0.28 | 0.17 | -0.03 | 0.63 |
| Income (30–60 thousand vs. below) | 1.78 | 0.83 | 2.14* | 0.68 | 0.45 | 1.5 | **0.25** | 0.13 | 0.02 | 0.52 |
| Income (10–30 thousand vs. below) | 0.65 | 0.89 | 0.73 | 0.009 | 0.48 | .02 | 0.09 | 0.13 | -0.15 | 0.34 |
| X*M (X = income) | | | | | | | $F(4, 815)$ = 0.9, $p$ = .5 | | | |
| Handedness (LQ) | 0.03 | .01 | 3.22** | -0.001 | 0.005 | -0.11 | **0.004** | 0.001 | 0.001 | 0.007 |
| X*M (X = Handedness) | | | | | | | $F(1, 818)$ = 0.00, $p$ = .98 | | | |
| Risk-taking (1vs 2) | -1.33 | 1.86 | -0.72 | -0.27 | 1.00 | -0.27 | -0.19 | 0.31 | -0.81 | 0.41 |
| Risk-taking (1vs 3) | -2.64 | 1.55 | -1.71 | -0.27 | 0.83 | -0.33 | -0.37 | 0.23 | -0.84 | 0.07 |
| Risk-taking (1vs 4) | -0.15 | 1.10 | -0.13 | 0.37 | 0.59 | 0.63 | -0.02 | 0.16 | -0.32 | 0.30 |
| Risk-taking (1vs 5) | -4.69 | 1.44 | -3.26** | 0.57 | 0.78 | 0.74 | **-0.67** | 0.22 | -1.1 | -0.26 |
| Risk-taking (1vs 6) | -0.49 | 1.43 | -0.34 | -0.40 | 0.77 | -0.52 | -0.07 | 0.22 | -0.50 | 0.38 |
| Risk-taking (1vs 7) | -0.11 | 0.94 | -0.12 | 1.67 | 0.50 | 3.32** | -0.02 | 0.14 | -0.29 | 0.26 |
| X*M (X = Risk-taking) | | | | | | | $F(6, 813)$ = 2.54, $p$ = .02 | | | |
| Age | 0.38 | 0.11 | 3.46*** | 0.09 | 0.06 | 1.54 | **0.05** | 0.02 | 0.02 | 0.09 |
| X*M (X = Age) | | | | | | | $F(1, 818)$ = 1.36, $p$ = .2 | | | |
| Sex | 0.89 | 0.68 | 1.32 | 1.02 | 0.36 | 2.80** | 0.13 | 0.1 | -0.06 | 0.33 |
| X*M (X = Sex) | | | | | | | $F(1, 818)$ = 0.93, $p$ = .3 | | | |
| Religion (Hindu vs. muslim) | -1.04 | 1.06 | -0.98 | 1.38 | 0.57 | 2.42* | -0.03 | 0.03 | -0.08 | 0.02 |
| Religion (Hindu vs. others) | 0.95 | 1.31 | 0.73 | 2.88 | 0.70 | 4.11** | 0.02 | 0.04 | -0.05 | 0.10 |
| X*M (X = Religion) | | | | | | | $F(2, 817)$ = 0.97, $p$ = .4 | | | |
| | $R^2$ = 0.07, $F(17, 820)$ = 3.7, $p$ < .001 | | | $R^2$ = 0.15, $F(18, 819)$ = 8.07, $p$ < .001 | | | | | | |

*Note.* M = mediation variable (intolerance of uncertainty), Y = dependent variable (believing), Coeff. = coefficient, SE = standard error, LCI = lower limit of 95% confidence interval, UCI = upper limit of 95% confidence interval, GC = general castes, OBC = other backward castes, SC = scheduled castes, LQ = laterality quotient, X*M = interaction of antecedent and mediation variables (moderation effect); bold values indicate significant indirect effects (not including zero between upper and lower limits of 95% confidence interval).

*** Significant at the .001 level.

** Significant at the .01 level.

* Significant at the .05 level.

**Table 5. The effects (direct and indirect effect) of antecedent and mediation (general IU) variables on bonding in the students' sample.**

| Antecedent variable | Effect on M | | | Direct effect on Y | | | Indirect effect on Y | | | |
|---|---|---|---|---|---|---|---|---|---|---|
| | Coeff. | SE | t | Coeff. | SE | t | Coeff. | SE | LCI | UCI |
| Intolerance of uncertainty | | | | .12 | .02 | 6.3*** | | | | |
| Caste (GC vs. others) | 1.69 | 0.76 | 2.23* | -0.39 | 0.41 | -0.96 | **0.20** | 0.09 | 0.03 | 0.39 |
| Caste (OBC vs.SC) | -0.23 | 0.83 | -0.28 | -2.01 | 0.45 | -4.5*** | -0.03 | 0.10 | -0.24 | 0.18 |
| X*M (X = caste) | | | | | | | $F(2, 817) = 1.9, p = .2$ | | | |
| Income (above vs. below 1lakh) | 1.03 | 1.69 | 0.61 | -0.53 | 0.90 | -0.59 | 0.12 | 0.21 | -0.29 | 0.56 |
| Income (60 thousand-1Lakh vs. below) | 1.97 | 1.17 | 1.69 | 1.58 | 0.63 | 2.53* | 0.23 | 0.14 | -0.04 | 0.52 |
| Income (30–60 thousand vs. below) | 1.78 | 0.83 | 2.14* | -0.49 | .44 | -1.09 | **0.21** | 0.10 | 0.01 | 0.42 |
| Income (10–30 thousand vs. below) | 0.65 | 0.89 | 0.73 | -0.90 | 0.48 | -1.88 | 0.08 | 0.10 | -0.12 | 0.29 |
| X*M (X = income) | | | | | | | $F(4, 815) = 1.1, p = .3$ | | | |
| Handedness (LQ) | 0.03 | .01 | 3.2** | 0.004 | 0.005 | 0.8 | **0.003** | 0.001 | 0.001 | 0.006 |
| X*M (X = Handedness) | | | | | | | $F(1, 818) = 0.04, p = .8$ | | | |
| Risk-taking (1vs 2) | -1.33 | 1.86 | -0.72 | -1.02 | 0.99 | -1.03 | -0.16 | 0.25 | -0.70 | 0.30 |
| Risk-taking (1vs 3) | -2.64 | 1.55 | -1.71 | 0.81 | 0.83 | 0.98 | -0.31 | 0.19 | -0.70 | 0.07 |
| Risk-taking (1vs 4) | -0.15 | 1.10 | -0.13 | 0.61 | 0.59 | 1.04 | -0.02 | 0.13 | -0.27 | 0.24 |
| Risk-taking (1vs 5) | -4.69 | 1.44 | -3.26** | -0.14 | 0.78 | -0.19 | **-0.55** | 0.19 | -0.95 | -0.20 |
| Risk-taking (1vs 6) | -0.49 | 1.43 | -0.34 | -0.73 | 0.77 | -0.96 | -0.06 | 0.18 | -0.41 | 0.30 |
| Risk-taking (1vs 7) | -0.11 | 0.94 | -0.12 | 1.50 | 0.50 | 2.98** | -0.01 | 0.12 | -0.24 | 0.22 |
| X*M (X = Risk-taking) | | | | | | | $F(6, 813) = 0.81, p = .6$ | | | |
| Age | 0.38 | 0.11 | 3.46*** | 0.02 | 0.06 | 0.29 | **0.04** | 0.02 | 0.02 | 0.08 |
| X*M (X = Age) | | | | | | | $F(1, 818) = 0.2, p = .7$ | | | |
| Sex | 0.89 | 0.68 | 1.32 | 1.26 | 0.36 | 3.5*** | 0.1 | 0.08 | -0.05 | 0.27 |
| X*M (X = Sex) | | | | | | | $F(1, 818) = 2.45, p = .12$ | | | |
| Religion (Hindu vs. muslim) | -1.04 | 1.06 | -0.98 | 0.83 | 0.57 | 1.46 | -0.12 | 0.12 | -0.37 | 0.10 |
| Religion (Hindu vs. others) | 0.95 | 1.31 | 0.73 | 2.31 | 0.7 | 3.3*** | 0.11 | 0.17 | -0.22 | 0.46 |
| X*M (X = Religion) | | | | | | | $F(2, 817) = 1.14, p = .3$ | | | |
| | $R^2 = 0.07, F(17, 820) = 3.7, p < .001$ | | | $R^2 = 0.13, F(18, 819) = 6.82, p < .001$ | | | | | | |

*Note*. M = mediation variable (intolerance of uncertainty), Y = dependent variable (bonding), Coeff. = coefficient, SE = standard error, LCI = lower limit of 95% confidence interval, UCI = upper limit of 95% confidence interval, GC = general castes, OBC = other backward castes, SC = scheduled castes, LQ = laterality quotient, X*M = interaction of antecedent and mediation variables (moderation effect); bold values indicate significant indirect effects (not including zero between upper and lower limits of 95% confidence interval).

*** Significant at the .001 level.

** Significant at the .01 level.

* Significant at the .05 level.

The indirect (mediated by general IU) effect of the lower caste status (GC vs. others) and the direct effect of the higher caste status (SC vs. OBC) were similar in believing, bonding, behaving, and belonging. The indirect effect of the lower family income (30–60 thousand vs. below) was similar in believing, bonding, behaving, and belonging, whereas the direct effect of the lower family income was similar in bonding and belonging, at one level (60 thousand-1 lakh vs. below), and similar in believing and behaving, at another level (above vs. below 1 lakh). The indirect effects of right-handedness and age and the direct effect of women were similar in believing, bonding, behaving, and belonging. The indirect effect of the intermediate level of risk-taking (scoring 5, compared to the lowest level) was similar in believing, bonding, behaving, and belonging, whereas the direct effect of the highest level of risk-taking (scoring 7, compared to the lowest level) was similar in believing, bonding, and behaving only (no effect

**Table 6. The effects (direct and indirect effect) of antecedent and mediation (general IU) variables on behaving in the students' sample.**

| Antecedent variable | Effect on M | | | Direct effect on Y | | | Indirect effect on Y | | | |
|---|---|---|---|---|---|---|---|---|---|---|
| | Coeff. | SE | $t$ | Coeff. | SE | $t$ | Coeff. | SE | LCI | UCI |
| Intolerance of uncertainty | | | | 0.11 | 0.02 | 5.6*** | | | | |
| Caste (GC vs. others) | 1.69 | 0.76 | 2.23* | -0.83 | 0.43 | -1.91 | **0.03** | 0.02 | 0.004 | 0.07 |
| Caste (OBC vs.SC) | -0.23 | 0.84 | -0.28 | -1.25 | 0.47 | -2.65** | -0.01 | .02 | -0.04 | 0.03 |
| X*M (X = caste) | | | | | | | $F(2, 817) = 0.3, p = .8$ | | | |
| Income (above vs. below 1lakh) | 1.04 | 1.69 | 0.61 | 2.37 | 0.96 | 2.48* | 0.11 | 0.20 | -0.28 | 0.52 |
| Income (60 thousand-1Lakh vs. below) | 1.97 | 1.17 | 1.69 | 0.50 | 0.66 | 0.75 | 0.22 | 0.14 | -0.03 | 0.53 |
| Income (30–60 thousand vs. below) | 1.78 | 0.83 | 2.14* | 0.18 | 0.47 | 0.37 | **0.20** | 0.10 | 0.01 | 0.41 |
| Income (10–30 thousand vs. below) | 0.65 | 0.89 | 0.73 | -0.58 | 0.50 | -1.16 | 0.07 | 0.10 | -0.11 | 0.28 |
| X*M (X = income) | | | | | | | $F(4, 815) = 0.81, p = .5$ | | | |
| Handedness (LQ) | 0.03 | .01 | 3.2** | -0.002 | 0.005 | -0.4 | **0.003** | 0.001 | 0.001 | 0.006 |
| X*M (X = Handedness) | | | | | | | $F(1, 818) = 2.02, p = .2$ | | | |
| Risk-taking (1vs 2) | -1.33 | 1.86 | -0.72 | -0.42 | 1.05 | -0.4 | -0.15 | 0.24 | -0.67 | 0.32 |
| Risk-taking (1vs 3) | -2.64 | 1.55 | -1.71 | 0.54 | 0.88 | 0.61 | -0.29 | 0.18 | -0.67 | .05 |
| Risk-taking (1vs 4) | -0.15 | 1.10 | -0.13 | 0.80 | 0.63 | 1.29 | -0.02 | 0.13 | -0.26 | 0.23 |
| Risk-taking (1vs 5) | -4.69 | 1.44 | -3.3** | 0.35 | 0.82 | 0.44 | **-0.52** | 0.18 | -0.9 | -0.19 |
| Risk-taking (1vs 6) | -0.49 | 1.43 | -0.34 | -0.14 | 0.81 | -0.18 | -0.05 | 0.17 | -0.4 | 0.28 |
| Risk-taking (1vs 7) | -0.11 | 0.94 | -0.12 | 1.65 | 0.53 | 3.1** | -0.01 | 0.11 | -0.22 | 0.21 |
| X*M (X = Risk-taking) | | | | | | | $F(6, 813) = 0.62, p = .7$ | | | |
| Age | 0.38 | 0.11 | 3.46*** | 0.07 | 0.06 | 1.1 | **0.04** | 0.02 | 0.01 | 0.08 |
| X*M (X = Age) | | | | | | | $F(1, 818) = 0.22, p = .6$ | | | |
| Sex | 0.89 | 0.68 | 1.32 | 1.03 | 0.38 | 2.7** | 0.10 | 0.08 | -0.05 | 0.26 |
| X*M (X = Sex) | | | | | | | $F(1, 818) = 0.63, p = .4$ | | | |
| Religion (Hindu vs. muslim) | -1.04 | 1.06 | -0.98 | 2.08 | 0.60 | 3.5*** | -0.02 | 0.02 | -0.06 | 0.01 |
| Religion (Hindu vs. others) | 0.95 | 1.31 | 0.73 | 3.77 | 0.74 | 5.1*** | 0.02 | 0.03 | -0.04 | 0.08 |
| X*M (X = Religion) | | | | | | | $F(2, 817) = 0.49, p = .6$ | | | |
| | $R^2 = 0.07, F(17, 820) = 3.66, p < .001$ | | | $R^2 = 0.13, F(18, 819) = 6.61, p < .001$ | | | | | | |

*Note*. M = mediation variable (intolerance of uncertainty), Y = dependent variable (behaving), Coeff. = coefficient, SE = standard error, LCI = lower limit of 95% confidence interval, UCI = upper limit of 95% confidence interval, GC = general castes, OBC = other backward castes, SC = scheduled castes, LQ = laterality quotient, X*M = interaction of antecedent and mediation variables (moderation effect); bold values indicate significant indirect effects (not including zero between upper and lower limits of 95% confidence interval).

*** Significant at the .001 level.

** Significant at the .01 level.

* Significant at the .05 level.

in belonging). Moreover, the direct effect of other-religion (compared to Hindu) was similar in believing, bonding, behaving, and belonging, whereas the direct effect of Muslim (compared to Hindu) was similar in believing, behaving, and belonging only (no effect in bonding, i.e., Hindus and Muslims were similar in bonding). There was no moderation effect (X-M interaction), except that risk-taking moderated the effect of IU on believing.

The indirect effect of height was similar in believing ($b$ = -0.80, SE = 0.37, 95% CI = -1.64 to -.20), bonding ($b$ = -0.63, SE = 0.29, 95% CI = -1.3 to -0.2), behaving ($b$ = -0.60, SE = 0.31, 95% CI = -1.3 to -0.11), and belonging ($b$ = -0.96, SE = 0.41, 95% CI = -1.9 to -.30), among women.

**The interrelationships in the dimensions of religiousness.** The inter-correlations in the dimensions of religiousness (see Table 8) among the whole sample varied from .50 to .62. The inter-correlations in the dimensions of religiousness were similar (the 'others' were not

**Table 7. The effects (direct and indirect effect) of antecedent and mediation (general IU) variables on belonging in the students' sample.**

| | Effect on M | | | Direct effect on Y | | | Indirect effect on Y | | | |
|---|---|---|---|---|---|---|---|---|---|---|
| Antecedent variable | Coeff. | SE | $t$ | Coeff. | SE | $t$ | Coeff. | SE | LCI | UCI |
| Intolerance of uncertainty | | | | 0.15 | 0.02 | 7.11*** | | | | |
| Caste (GC vs. others) | 1.69 | 0.76 | 2.23* | 0.17 | 0.46 | 0.38 | **0.25** | 0.12 | 0.03 | 0.48 |
| Caste (OBC vs.SC) | -0.23 | 0.83 | -0.28 | -0.73 | 0.50 | -1.47 | -0.03 | 0.13 | -0.31 | 0.21 |
| X*M (X = caste) | | | | | | | $F(2, 817) = 0.51, p = .6$ | | | |
| Income (above vs. below 1lakh) | 1.03 | 1.69 | 0.61 | 0.53 | 1.01 | 0.53 | 0.15 | 0.26 | -0.37 | 0.67 |
| Income (60 thousand-1Lakh vs. below) | 1.97 | 1.17 | 1.69 | 1.47 | 0.70 | 2.10* | 0.29 | 0.18 | -0.04 | 0.68 |
| Income (30–60 thousand vs. below) | 1.78 | 0.83 | 2.14* | -0.55 | 0.50 | -1.11 | **0.26** | 0.13 | 0.01 | 0.53 |
| Income (10–30 thousand vs. below) | 0.65 | 0.89 | 0.73 | -0.27 | 0.53 | -0.50 | 0.10 | 0.13 | -0.16 | 0.36 |
| X*M (X = income) | | | | | | | $F(4, 815) = 0.84, p = .5$ | | | |
| Handedness (LQ) | 0.03 | 0.01 | 3.22** | 0.002 | 0.005 | 0.45 | **0.004** | 0.002 | 0.002 | .007 |
| X*M (X = Handedness) | | | | | | | $F(1, 818) = 0.24, p = .6$ | | | |
| Risk-taking (1vs 2) | -1.33 | 1.86 | -0.72 | 0.59 | 1.11 | 0.53 | -0.20 | 0.32 | -0.89 | 0.40 |
| Risk-taking (1vs 3) | -2.64 | 1.55 | -1.71 | 0.57 | 0.93 | 0.62 | -0.39 | 0.24 | -0.87 | 0.07 |
| Risk-taking (1vs 4) | -0.15 | 1.10 | -0.13 | 0.28 | 0.66 | 0.42 | -0.02 | 0.17 | -0.35 | 0.30 |
| Risk-taking (1vs 5) | -4.69 | 1.44 | -3.26** | -1.16 | 0.87 | -1.33 | **-0.70** | 0.23 | -1.18 | -0.28 |
| Risk-taking (1vs 6) | -0.49 | 1.43 | -0.34 | -0.15 | 0.86 | -0.18 | -0.07 | 0.22 | -0.51 | 0.36 |
| Risk-taking (1vs 7) | -0.11 | 0.94 | -0.12 | 1.09 | 0.56 | 1.94 | -0.02 | 0.14 | -0.31 | 0.27 |
| X*M (X = Risk-taking) | | | | | | | $F(6, 813) = 1.38, p = .2$ | | | |
| Age | 0.38 | 0.11 | 3.46*** | 0.16 | 0.07 | 2.37* | **0.06** | 0.02 | 0.02 | 0.10 |
| X*M (X = Age) | | | | | | | $F(1, 818) = 1.24, p = .3$ | | | |
| Sex | 0.89 | 0.68 | 1.32 | 0.93 | 0.40 | 2.29* | 0.13 | 0.10 | -0.07 | 0.34 |
| X*M (X = Sex) | | | | | | | $F(1, 818) = 0.00, p = .96$ | | | |
| Religion (Hindu vs. muslim) | -1.04 | 1.06 | -0.98 | 1.50 | 0.63 | 2.37* | -0.15 | 0.15 | -0.44 | 0.13 |
| Religion (Hindu vs. others) | 0.95 | 1.30 | 0.73 | 2.50 | 0.78 | 3.20** | 0.14 | 0.22 | -0.28 | 0.59 |
| X*M (X = Religion) | | | | | | | $F(2, 817) = 0.51, p = .6$ | | | |
| | $R^2 = 0.07, F(17, 820) = 3.66, p < .001$ | | | $R^2 = 0.12, F(18, 819) = 6.11, p < .001$ | | | | | | |

*Note.* M = mediation variable (intolerance of uncertainty), Y = dependent variable (belonging), Coeff. = coefficient, SE = standard error, LCI = lower limit of 95% confidence interval, UCI = upper limit of 95% confidence interval, GC = general castes, OBC = other backward castes, SC = scheduled castes, LQ = laterality quotient, X*M = interaction of antecedent and mediation variables (moderation effect); bold values indicate significant indirect effects (not including zero between upper and lower limits of 95% confidence interval).

*** Significant at the .001 level.

** Significant at the .01 level.

* Significant at the .05 level.

analyzed because of the mixed religion category) among Hindus (.51 to .61), and Muslims (.34 to .60). The inter-correlations of the dimensions of religiousness were slightly higher among SC (.58 to .68) than among GC (.50 to .62) and OBC (.44 to .57). The inter-correlations in the dimensions of religiousness were slightly higher among participants with income below 30 thousand (.54 to .65) than with income 30 to 60 thousand (.40 to .56), and income above 60 thousand (.48 to .57). Interestingly, for all these groups, after controlling religiousness, the partial inter-correlations were negative and, mostly, significant. This shows that the dimensions of religiousness were distinct from the general religiousness.

**The relative importance of the dimensions of religiousness in different groups.** We conducted one-way repeated measures analyses of the dimensions of religiousness for different groups (see Table 9). Among Hindus, bonding was highest, followed by believing, which, in

**Table 8. Simple and partial (in parenthesis; after controlling religiousness) correlations between the dimensions of religiousness in social, economic, and religion groups among the students' sample.**

| Students | Bel.-Bond. | Bel.-Beh. | Bel.-Blg. | Bond.-Beh. | Bond.-Blg. | Beh.-Blg. |
|---|---|---|---|---|---|---|
| Total (*n* = 861) | .56 [.50, .61] (-.29 [-.37, -.21]) | .62 [.56, .67] (-.26 [-.34, -.18]) | .52 [.47, .58] (-.40 [-.46, -.33]) | .59 [.53, .64] (-.30 [-.38, -.23]) | .50 [.44, .56] (-.39 [-.46, -.32]) | .57 [.52, .63] (-.35 [-.41, -.26]) |
| Hindu (*n* = 698) | .56 [.51, .62] (-.28 [-.36, -.20]) | .60 [.54, .65] (-.30 [-.38, -.22]) | .52 [.46, .58] (-.39 [-.46, -.32]) | .61 [.55, .66] (-.27 [-.35, -.18]) | .56 [.51, .62] (-.42 [-.49, -.33]) | .58 [.52, .64] (-.33 [-.41, -.25]) |
| Muslim (*n* = 98) | .46 [.25, .65] (-.25 [-.50, -.00]) | .60 [.42, .75] (-.10 [-.31, .14]) | .44 [.21, .60] (-.49 [-.63, -.30]) | .34 [.08, .57] (-.46 [-.36, -.20]) | .39 [.15, .58] (-.30 [-.60, -.28]) | .45 [.25, .64] (-.28 [-.59, -.12]) |
| GC (*n* = 235) | .50 [.39, .60] (-.39 [-.49, -.25]) | .62 [.52, .71] (-.21 [-.35, -.07]) | .54 [.42, .64] (-.38 [-.51, -.24]) | .60 [.49, .70] (-.25 [-.41, -.08]) | .54 [.42, .65] (-.36 [-.50, -.20]) | .58 [.48, .68] (-.40 [-.51, -.26]) |
| OBC (*n* = 385) | .49 [.41, .57] (-.26 [-.37, -.13]) | .57 [.49, .65] (-.26 [-.38, -.14]) | .44 [.34, .52] (-.44 [-.53, -.33]) | .55 [.45, .63] (-.29 [-.40, -.18]) | .43 [.33, .53] (-.40 [-.49, -.29]) | .54 [.45, .63] (-.33 [-.44, -.21]) |
| SC (*n* = 234) | .66 [.58, .73] (-.25 [-.40, -.11]) | .68 [.58, .77] (-.28 [-.41, -.10]) | .62 [.51, .71] (-.40 [-.52, -.27]) | .62 [.52, .71] (-.39 [-.49, -.29]) | .58 [.48, .68] (-.39 [-.50, -.25]) | .64 [.55, .73] (-.29 [-.42, -.15]) |
| Income 1 (*n* = 111) | .49 [.33, .62] (-.35 [-.52, -.13]) | .57 [.42, .69] (-.28 [-.47, -.07]) | .48 [.32, .63] (-.42 [-.58, -.24]) | .55 [.39, .70] (-.30 [-.52, -.06]) | .49 [.30, .65] (-.36 [-.56, -.15]) | .57 [.41, .71] (-.28 [-.48, -.06]) |
| Income 2 (*n* = 191) | .45 [.32, .58] (-.31 [-.47, -.14]) | .56 [.43, .67] (-.29 [-.46, -.10]) | .44 [.31, .56] (-.39 [-.52, -.25]) | .51 [.38, .63] (-.30 [-.45, -.15]) | .40 [.26, .54] (-.37 [-.51, -.19]) | .51 [.39, .62] (-.34 [-.49, -.17]) |
| Income 3 (*n* = 556) | .61 [.54, .67] (-.26 [-.36, -.17]) | .65 [.59, .71] (-.25 [-.33, -.17]) | .56 [.49, .62] (-.39 [-.47, -.31]) | .62 [.56, .68] (-.30 [-.38, -.21]) | .54 [.46, .60] (-.42 [-.49, -.33]) | .59 [.53, .66] (-.37 [-.45, -.28]) |

*Note*. Bel. = believing, Bond. = bonding, Beh. = behaving, Blg. = belonging, GC = general castes, OBC = other backward castes, SC = scheduled castes, Income 1 = above 60 thousand, Income 2 = 30–60 thousand, Income 3 = below 30 thousand. For a more meaningful understanding (aligned with path analysis), in this analysis, original five categories were merged to form three categories of income. One thousand bootstrap cycles, for estimation of 95% confidence interval, were used to derive correlations. Brackets report 95% confidence interval.

turn, was higher than behaving (belonging is lower than bonding). Among Muslims, there was no difference in the dimensions of religiousness. However, bonding tends to be the highest (in mean scores). Among GC, bonding was higher than all other dimensions. Among OBC, bonding was highest, followed by believing, which, in turn, was higher than behaving (belonging is lower than bonding). In SC, behaving was lower than all other dimensions. Moreover, among the higher-income group (above 60 thousand), there was no difference in the dimensions of religiousness, whereas, among the intermediate-income group (30–60 thousand), bonding was highest, followed by believing, which, in turn, was higher than behaving (belonging is lower

**Table 9. The distribution and the comparison of the dimensions of religiousness in different groups among the students sample.**

| | Believing[a] | Bonding[b] | Behaving[c] | Belonging[d] | Comparisons | | | |
|---|---|---|---|---|---|---|---|---|
| | *M* (*SD*) | *M* (*SD*) | *M* (*SD*) | *M* (*SD*) | *df* | *F* | η² | |
| Hindu | 5.08 (0.07)[b,c] | 5.36 (0.07)[a,c,d] | 4.86 (0.07)[a,b] | 4.99 (0.08)[b] | 3, 695 | 21.5*** | .09 | b > a > c; b > d |
| Muslims | 5.52 (0.17) | 5.75 (0.15) | 5.6 (0.17) | 5.5 (0.18) | 3, 95 | 0.9 | .03 | |
| GC | 5.09 (0.12)[b] | 5.47 (0.12)[a,c,d] | 5.22 (0.12)[b] | 4.99 (0.13)[b] | 3, 232 | 6*** | .07 | b > a, c, d |
| OBC | 5.36 (0.09)[b,c] | 5.69 (0.08)[a,c,d] | 5.12 (0.09)[a,b] | 5.21 (0.10)[b] | 3, 382 | 16*** | .11 | b > a > c; b > d |
| SC | 5.06 (0.13)[c] | 5.04 (0.13)[c] | 4.75 (0.13)[a,b,d] | 5.03 (0.13)[c] | 3, 231 | 3.6* | .05 | a, b, d > c |
| Income 1 | 4.78 (0.19) | 5.11 (0.18) | 4.7 (0.18) | 4.67 (0.18) | 3, 108 | 2.35 | .06 | |
| Income 2 | 5.04 (0.13)[b] | 5.5 (0.12)[a,c,d] | 4.94 (0.14)[b,d] | 5.22 (0.13) | 3, 188 | 7.3*** | .10 | b > a > c; b > d |
| Income 3 | 5.35 (0.08)[b,c,d] | 5.5 (0.08)[a,c,d] | 5.15 (0.08)[a,b] | 5.15 (0.09)[a,b] | 3, 553 | 11.1*** | .06 | b > a > c, d |

*Note*. Superscripted letters indicate differences between the dimensions of religiousness, significant at the .05 level. GC = general castes, OBC = other backward castes, SC = scheduled castes, Income 1 = above 60 thousand, Income 2 = 30–60 thousand, Income 3 = below 30 thousand. For a more meaningful understanding, in this analysis, original five categories were merged to form three categories of income.

than bonding) and, among the lower-income group (below 30 thousand), bonding was highest, followed by believing, which, in turn, was higher than behaving and belonging.

## Community sample

Table 10 shows the distribution of variables. After the list-wise deletion of missing cases, path analyses were conducted on a sample of 232 cases.

**General IU as the mediator of the effects of antecedent variables on general religiousness.** Fig 2 shows the final path analysis model of the effect of SES and other factors on religiousness. Table 11 shows the effects of antecedent variables on mediation and (direct and indirect effects) dependent variables.

There were significant indirect effects (60 thousand-1lakh vs. lower income; 30–60 thousand vs. lower income), but no direct effects (omnibus test, $F(4, 216) = 0.42$, $p = .79$), of family

**Table 10. The distribution of variables in the community sample.**

| Variable | M | SD | Range | N |
|---|---|---|---|---|
| Religiousness | 5.60 | 1.29 | 1.83–7 | 247 |
| Believing | 5.66 | 1.57 | 1–7 | 249 |
| Bonding | 5.70 | 1.66 | 1–7 | 249 |
| Behaving | 5.45 | 1.80 | 1–7 | 247 |
| Belonging | 5.59 | 1.79 | 1–7 | 250 |
| IU | 39.50 | 8.83 | 12–60 | 249 |
| Prospective IU | 24.73 | 5.40 | 7–35 | 249 |
| Inhibitive IU | 14.75 | 4.65 | 5–25 | 250 |
| Age | 37.55 | 6.95 | 27–63 | 250 |
| Sex (female) | 0.37 | 0.49 | 0–1 | 248 |
| Handedness (LQ) | 76.90 | 42.89 | -100- 100 | 246 |
| Height (in meters) | 1.64 | 0.08 | 1.42–1.85 | 248 |
| BMI | 39.36 | 5.80 | 25.40–62.13 | 247 |
| Caste status (OBC) | 0.43 | 0.50 | 0–1 | 245 |
| Caste status (SC) | 0.11 | 0.31 | 0–1 | 245 |
| Income (60 thousand-1lakh) | 0.11 | 0.31 | 0–1 | 248 |
| Income (30–60 thousand) | 0.28 | 0.45 | 0–1 | 248 |
| Income (10–30 thousand) | 0.33 | 0.47 | 0–1 | 248 |
| Income (< 10 thousand) | 0.19 | 0.39 | 0–1 | 248 |
| Religion (muslim) | 0.09 | 0.28 | 0–1 | 250 |
| Religion (others) | 0.04 | 0.20 | 0–1 | 250 |
| Risktaking score (2) | 0.04 | 0.19 | 0–1 | 249 |
| Risktaking score (3) | 0.06 | 0.23 | 0–1 | 249 |
| Risktaking score (4) | 0.17 | 0.38 | 0–1 | 249 |
| Risktaking score (5) | 0.05 | 0.21 | 0–1 | 249 |
| Risktaking score (6) | 0.08 | 0.28 | 0–1 | 249 |
| Risktaking score (7) | 0.31 | 0.46 | 0–1 | 249 |
| Education (UG) | 0.35 | 0.48 | 0–1 | 248 |
| Education (PG) | 0.34 | 0.48 | 0–1 | 248 |
| Education (higher) | 0.06 | 0.25 | 0–1 | 248 |

*Note*. IU = intolerance of uncertainty, LQ = laterality quotient, BMI = body mass index, OBC = other backward castes, SC = scheduled castes, UG = under-graduation PG = post-graduation. *N* shows the number of participants after deletion of missing cases.

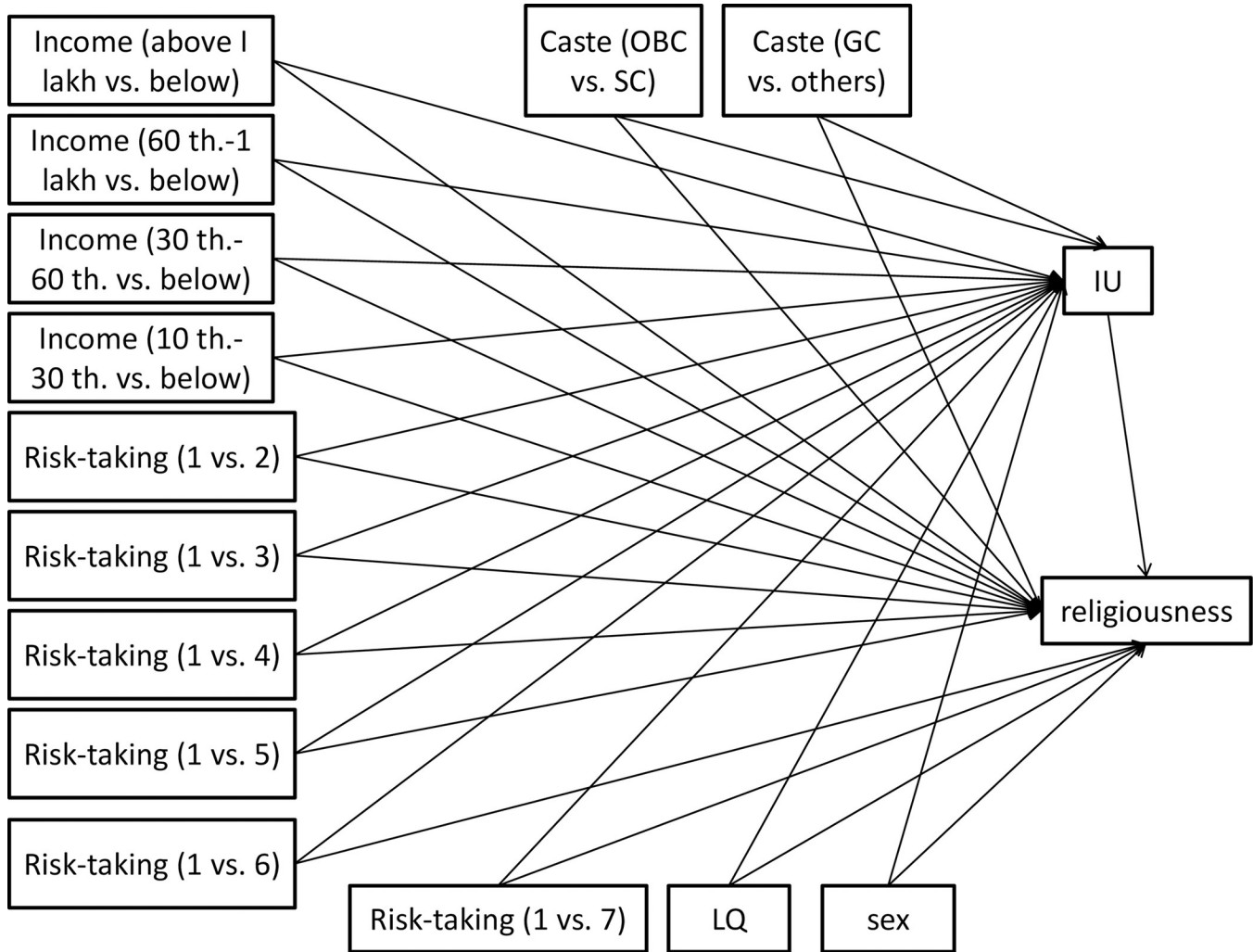

**Fig 2. The conceptual path diagram shows the studied relationships of antecedent variables with IU and religiousness in the community sample.** IU = intolerance of uncertainty, GC = general castes, OBC = other backward castes, SC = scheduled castes, th. = thousand, LQ = laterality quotient (handedness).

income on religiousness. Thus, a decrease in family income leads to an increase in IU, which, in turn, leads to an increase in religiousness.

The omnibus direct effect of caste status on religiousness was (nominally) not significant ($F(2, 216) = 3.01$, $p = .051$). However, the relative direct effect of caste status (GC vs. others) on religiousness was significant. There was no indirect effect of caste status on religiousness. Thus, an increase in caste status (higher status) leads to a direct increase in religiousness.

Although omnibus direct effect was not significant ($F(6, 216) = 1.8$, $p = .10$), relative direct effects of risk-taking (score 1 vs. 2; score 1 vs. 4) on religiousness were significant. Thus, intermediate-level risk-takers (compared to lowest-level risk-takers) have lower religiousness. Moreover, there was also an indirect effect of risk-taking on religiousness (score 1 vs. 3). Thus, intermediate-level risk-takers (scoring 3, compared to lowest-level risk-takers) have lower IU, which, in turn, leads to lower religiousness.

Moreover, sex had a direct effect, but no indirect effect, on religiousness. Thus, women have higher religiousness than men. Similarly, handedness had a direct effect, but no indirect effect, on religiousness. Thus, right-handers have higher religiousness than left-handers. There

**Table 11. The effects (direct and indirect effects) of antecedent variables on the mediation variable (general IU) and the dependent variable (religiousness) in the community sample.**

| Antecedent | Effect on M | | | Direct effect on Y | | | Indirect effect on Y | | | |
|---|---|---|---|---|---|---|---|---|---|---|
| | Coeff. | SE | *t* | Coeff. | SE | *t* | Coeff. | SE | LCI | HCI |
| Intolerance of uncertainty | | | | 0.42 | 0.12 | 3.6*** | | | | |
| Caste (GC vs. others) | 1.35 | 1.34 | 1.0 | -5.7 | 2.3 | -2.5* | 0.56 | 0.59 | -0.6 | 1.8 |
| Caste (OBC vs.SC) | 0.80 | 1.98 | 0.4 | -4.4 | 3.4 | -1.3 | 0.34 | 0.88 | -1.4 | 2.1 |
| X*M (X = Caste) | | | | | | | *F* (2, 214) = 1.1, *p* = .3 | | | |
| Income (above vs. below 1lakh) | 1.3 | 1.93 | 0.7 | 3.94 | 3.33 | 1.2 | 0.54 | 0.87 | -1.3 | 2.3 |
| Income (60 thousand-1lakh vs. below) | 3.88 | 1.9 | 2.0* | 0.78 | 3.31 | 0.2 | **1.63** | 0.84 | 0.2 | 3.4 |
| Income (30–60 thousand vs. below) | 4.12 | 1.4 | 2.9** | 0.75 | 2.47 | 0.3 | **1.73** | 0.76 | 0.4 | 3.3 |
| Income (10–30 thousand vs. below) | -0.35 | 1.7 | -0.2 | -0.5 | 2.94 | -0.2 | -0.15 | 0.77 | -1.7 | 1.4 |
| X*M (X = Income) | | | | | | | *F* (4, 212) = 1.3, *p* = .3 | | | |
| Handedness | 0.0005 | .01 | 0.04 | 0.05 | 0.03 | 2.0* | 0.0002 | 0.005 | -0.01 | 0.01 |
| X*M (X = Handedness) | | | | | | | *F* (1, 215) = 2.0, *p* = .2 | | | |
| Sex | -0.95 | 1.24 | -0.8 | 4.24 | 2.14 | 2.0* | -0.4 | 0.53 | -1.6 | 0.5 |
| X*M (X = Sex) | | | | | | | *F* (1, 215) = 1.5, *p* = .2 | | | |
| Risk-taking (1vs 2) | -5.2 | 3.31 | -1.6 | -12.04 | 5.74 | -2.1* | -2.19 | 1.46 | -5.5 | 0.1 |
| Risk-taking (1vs 3) | -5.1 | 2.71 | -1.9 | 1.58 | 4.71 | 0.3 | **-2.13** | 1.12 | -4.6 | -0.3 |
| Risk-taking (1vs 4) | -1.2 | 1.83 | -0.7 | -6.24 | 3.17 | -2.0* | -0.51 | 0.80 | -2.1 | 1.1 |
| Risk-taking (1vs 5) | -4.3 | 2.73 | -1.6 | 1.82 | 4.74 | 0.4 | -1.79 | 1.12 | -4.2 | 0.2 |
| Risk-taking (1vs 6) | -3.81 | 2.20 | -1.8 | 1.78 | 3.79 | 0.5 | -1.6 | 1.11 | -3.9 | 0.6 |
| Risk-taking (1vs 7) | -1.66 | 1.51 | -1.1 | -0.06 | 2.61 | -0.02 | -0.69 | 0.67 | -2.1 | 0.6 |
| X*M (X = Risk-taking) | | | | | | | *F* (6, 210) = 1.1, *p* = .4 | | | |
| | $R^2$ = 0.10, *F*(14, 217) = 1.7, *p* = .06 | | | $R^2$ = 0.16, *F*(15, 216) = 2.72, *p* < .001 | | | | | | |

Note. M = mediation variable (intolerance of uncertainty), Y = Dependent variable (religiousness), GC = general castes, OBC = other backward castes, SC = scheduled castes, LQ = laterality quotient, Coeff. = coefficient, SE = standard error, LCI = lower limit of 95% confidence interval, UCI = upper limit of 95% confidence interval, X*M = interaction of antecedent and mediation variables (moderation effect); bold values indicate significant indirect effects (not including zero between upper and lower limits of 95% confidence interval).

*** Significant at the .001 level.

** Significant at the .01 level.

* Significant at the .05 level.

was no moderation effect (X-M interaction) in any of the above-reported relationships (Table 11).

Age (indirect effect: *b* = -0.001, SE = 0.003, 95% CI = -0.007 to 0.004; direct effect: *b* = 0.009, *t*(229) = 0.8, *p* = .4), height (indirect effect: *b* = 0.18, SE = 0.29, 95% CI = -0.33 to 0.83; direct effect: *b* = -1.5, *t*(228) = -1.44, *p* = .15), BMI (indirect effect: *b* = -0.005, SE = 0.004, 95% CI = -0.013 to 0.002; direct effect: *b* = 0.02, *t*(227) = 1.52, *p* = .13), religion (indirect effects, Muslim (vs. Hindu): *b* = -0.07, SE = 0.09, 95% CI = -0.3 to 0.07; Other religion (vs. Hindu): *b* = -0.009, SE = 0.12, 95% CI = -0.26 to 0.22; omnibus direct effect: *F* (2, 228) = 1.97, *p* = .14) and education (indirect effect, undergraduate (vs. intermediate): *b* = -0.08, SE = 0.06, 95% CI = -0.2 to 0.02, postgraduate (vs. intermediate): *b* = -0.07, SE = 0.6, 95% CI = -0.2 to 0.03, Higher (vs. intermediate): *b* = -0.15, SE = 0.09, 95% CI = -0.4 to 0.001; omnibus direct effect: *F* (3, 226) = 0.83, *p* = .5) had no effects on religiousness. Therefore, these variables were excluded from the final model.

**Prospective and inhibitive IUs as mediators of the effects of antecedent variables on general religiousness.** There were no mediation effects of the combination of prospective

**Table 12. The effects (direct and indirect effect) of antecedent and mediation (IUP & IUI) variables on religiousness in the community sample.**

| | Effect on M1 | | | Effect on M2 | | | Direct effect on Y | | | Indirect effect (on Y) mediated by M1 | | | | Indirect effect (on Y) mediated by M2 | | | |
|---|---|---|---|---|---|---|---|---|---|---|---|---|---|---|---|---|---|
| | Cof. | SE | t | Cof. | SE | T | Cof. | SE | t | Cof. | SE | LCI | UCI | Cof. | SE | LCI | UCI |
| IUP | | | | | | | 0.37 | 0.23 | 1.6 | | | | | | | | |
| IUI | | | | | | | 0.49 | 0.27 | 1.8 | | | | | | | | |
| C1 | 0.67 | 0.83 | 0.8 | 0.68 | 0.69 | 1.0 | -5.7 | 2.3 | -2.5** | 0.25 | 0.41 | -0.5 | 1.2 | 0.33 | 0.41 | -0.4 | 1.23 |
| C2 | -0.35 | 1.23 | -0.3 | 1.15 | 1.02 | 1.1 | -4.5 | 3.44 | -1.3 | -0.13 | 0.57 | -1.4 | 1.08 | 0.56 | 0.61 | -0.6 | 1.88 |
| X*M | | | | | | | | | | $F(2, 213) = 1.3, p = .3$ | | | | $F(2, 213) = 0.3, p = .8$ | | | |
| I1 | 0.45 | 1.20 | 0.4 | 0.85 | 0.99 | 0.9 | 3.90 | 3.34 | 1.2 | 0.16 | 0.57 | -0.7 | 1.61 | 0.41 | 0.61 | -0.8 | 1.72 |
| I2 | -0.11 | 1.18 | -0.1 | 3.98 | 0.97 | 4.1*** | 0.50 | 3.46 | 0.2 | -0.04 | 0.50 | -1.1 | 1.08 | 1.94 | 1.15 | -0.1 | 4.39 |
| I3 | 1.32 | 0.87 | 1.5 | 2.80 | 0.72 | 3.9*** | 0.63 | 2.51 | 0.3 | 0.48 | 0.50 | -.03 | 1.63 | 1.36 | 0.85 | -0.1 | 3.22 |
| I4 | -0.87 | 1.06 | -0.8 | 0.52 | 0.88 | 0.60 | -0.58 | 2.97 | -0.2 | -0.32 | 0.49 | -1.5 | .054 | 0.25 | 0.54 | -0.7 | 1.50 |
| X*M | | | | | | | | | | $F(4, 211) = 1.7, p = .15$ | | | | $F(4, 211) = 0.5, p = .7$ | | | |
| LQ | -0.002 | 0.009 | -0.2 | 0.002 | 0.007 | 0.3 | 0.052 | 0.025 | 2.1* | -0.001 | 0.003 | -0.01 | 0.01 | 0.001 | 0.003 | -0.01 | 0.01 |
| X*M | | | | | | | | | | $F(1, 214) = 0.7, p = .4$ | | | | $F(1, 214) = 2.7, p = .1$ | | | |
| RT1 | -3.42 | 2.05 | -1.7 | -1.80 | 1.70 | -1.06 | -12.1 | 5.76 | -2.1* | -1.26 | 1.27 | -4.5 | 0.40 | -0.9 | 0.98 | -3.2 | 0.66 |
| RT2 | -3.43 | 1.68 | -2.0* | -1.65 | 1.39 | -1.19 | 1.52 | 4.73 | 0.3 | -1.26 | 1.01 | -3.6 | 0.33 | -0.8 | 0.81 | -2.8 | 0.38 |
| RT3 | -0.33 | 1.14 | -0.3 | -0.88 | 0.94 | -0.94 | -6.20 | 3.18 | -2.0 | -0.12 | 0.50 | -1.2 | 0.91 | -0.4 | 0.58 | -1.9 | 0.47 |
| RT4 | -2.36 | 1.70 | -1.4 | -1.90 | 1.40 | -1.36 | 1.82 | 4.75 | 0.4 | -0.87 | 0.94 | -3.1 | 0.55 | -0.9 | 0.76 | -2.7 | 0.24 |
| RT5 | -1.97 | 1.35 | -1.5 | -1.85 | 1.12 | -1.65 | 1.80 | 3.8 | 0.5 | -0.72 | 0.76 | -2.5 | 0.52 | -0.9 | 0.87 | -3.0 | 0.35 |
| RT6 | -0.28 | 0.94 | -0.3 | -1.37 | 0.77 | -1.78 | .019 | 2.63 | 0.0 | -0.10 | 0.42 | -1.1 | 0.72 | -0.7 | 0.56 | -2.0 | 0.19 |
| X*M | | | | | | | | | | $F(6, 209) = 1.4, p = .2$ | | | | $F(6, 209) = 1.0, p = .4$ | | | |
| Sex | 0.29 | 0.77 | 0.4 | -1.24 | 0.64 | -1.95 | 4.34 | 2.18 | 2.0* | 0.11 | 0.34 | -0.5 | 0.88 | -0.6 | 0.50 | -1.8 | 0.08 |
| X*M | | | | | | | | | | $F(1, 214) = 0.6, p = .4$ | | | | $F(1, 214) = 2.0, p = .2$ | | | |
| | $R^2 = 0.07, F(14, 217) = 1.07, p = .40$ | | | $R^2 = 0.17, F(14, 217) = 3.11, p < .001$ | | | $R^2 = 0.16, F(16, 215) = 2.54, p < .01$ | | | | | | | | | | |

*Note.* M1 = mediation variable 1 (IUP), M2 = mediation variable 2 (IUI), Y = Dependent variable (religiousness), Cof. = coefficient, SE = standard error, LCI = lower limit of 95% confidence interval, UCI = upper limit of 95% confidence interval, IUP = prospective intolerance of uncertainty, IUI = inhibitive intolerance of uncertainty, C1 = general castes vs. others, C2 = other backward castes vs. scheduled castes, I1 = Income (above vs. below 1lakh), I2 = Income (60 thousand-1Lakh vs. below), I3 = Income (30–60 thousand vs. below), I4 = Income (10–30 thousand vs. below), LQ = laterality quotient, RT1 = Risk-taking (1vs. 2), RT2 = Risk-taking (1vs. 3), RT3 = Risk-taking (1vs. 4), RT4 = Risk-taking (1vs. 5), RT5 = Risk-taking (1vs. 6), RT6 = Risk-taking (1vs. 7), X*M = interaction of antecedent (given in above row) and mediation variables (moderation effect); bold values indicate significant indirect effects (not including zero between upper and lower limits of 95% confidence interval).

*** Significant at the .001 level.

** Significant at the .01 level.

* Significant at the .05 level.

and inhibitive IUs. Therefore, we have reported the results of a parallel mediation model (see Table 12 and S3 Fig).

The lower family income (60 thousand-1lakh vs. lower income; 30–60 thousand vs. lower income) was related to inhibitive IU; however, it failed to translate into an (indirect) effect on religiousness. The intermediate level of risk-taking (scoring 3, compared to lowest-level risk-takers) was related to prospective IU; however, it failed to translate into an (indirect) effect on religiousness. Moreover, the intermediate level of risk-taking (scoring 2, compared to lowest-level risk-takers), the higher caste status (GC vs. others), right-handedness, and women (compared to men) were directly related to religiousness.

**General IU as the mediator of the effects of antecedent variables on the dimensions of religiousness.** The effects of antecedent variables on the dimensions of religiousness were largely similar to the effects reported for the general religiousness above (see Tables 13–16).

**Table 13. The effects (direct and indirect effect) of antecedent and mediation (general IU) variables on believing in the community sample.**

| Antecedent variable | Effect on M | | | Direct effect on Y | | | Indirect effect on Y | | | |
|---|---|---|---|---|---|---|---|---|---|---|
| | Coeff. | SE | t | Coeff. | SE | t | Coeff. | SE | LCI | UCI |
| Intolerance of uncertainty | | | | 0.14 | 0.04 | 3.34** | | | | |
| Caste (GC vs. others) | 1.35 | 1.34 | 1.00 | -2.05 | 0.84 | -2.44* | 0.19 | 0.20 | -0.20 | 0.60 |
| Caste (OBC vs.SC) | 0.80 | 1.98 | 0.41 | -1.96 | 1.24 | -1.58 | 0.11 | 0.30 | -0.51 | 0.71 |
| X*M (X = caste) | | | | | | | $F(2, 214) = 0.14, p = .9$ | | | |
| Income (above vs. below 1lakh) | 1.30 | 1.93 | 0.67 | 0.35 | 1.00 | 0.35 | 0.15 | 0.25 | -0.36 | 0.66 |
| Income (60 thousand-1Lakh vs. below) | 3.88 | 1.9 | 2.04* | -0.02 | 1.00 | -0.02 | **0.45** | 0.23 | 0.06 | 0.96 |
| Income (30–60 thousand vs. below) | 4.12 | 1.4016 | 2.94** | -1.16 | 0.74 | -1.56 | **0.48** | 0.22 | 0.11 | 0.97 |
| Income (10–30 thousand vs. below) | -0.35 | 1.71 | -0.21 | -0.50 | 0.89 | -0.56 | -0.04 | 0.22 | -0.49 | 0.37 |
| X*M (X = income) | | | | | | | $F(4, 212) = 2.34, p = .06$ | | | |
| Handedness (LQ) | 0.001 | 0.014 | 0.04 | 0.02 | 0.01 | 2.51* | 0.000 | .001 | -0.003 | .003 |
| X*M (X = Handedness) | | | | | | | $F(1, 215) = 4.08, p = .04$ | | | |
| Risk-taking (1vs 2) | -5.23 | 3.31 | -1.58 | -2.1 | 1.73 | -1.21 | -0.61 | 0.41 | -1.56 | 0.03 |
| Risk-taking (1vs 3) | -5.08 | 2.71 | -1.87 | 0.40 | 1.42 | 0.28 | **-0.59** | 0.32 | -1.30 | -0.07 |
| Risk-taking (1vs 4) | -1.22 | 1.83 | -0.66 | -1.79 | 0.95 | -1.87 | -0.14 | 0.22 | -0.58 | 0.31 |
| Risk-taking (1vs 5) | -4.26 | 2.73 | -1.56 | -0.59 | 1.43 | -0.42 | -0.49 | 0.32 | -1.18 | 0.08 |
| Risk-taking (1vs 6) | -3.82 | 2.18 | -1.75 | 0.59 | 1.14 | 0.52 | -0.44 | 0.32 | -1.13 | 0.15 |
| Risk-taking (1vs 7) | -1.66 | 1.51 | -1.1 | 0.15 | 0.79 | 0.19 | -0.19 | 0.19 | -0.60 | 0.18 |
| X*M (X = Risk-taking) | | | | | | | $F(6, 210) = 0.73, p = .6$ | | | |
| Sex | -0.95 | 1.24 | -0.76 | 0.74 | 0.65 | 1.15 | -0.11 | 0.15 | -0.44 | 0.17 |
| X*M (X = Sex) | | | | | | | $F(1, 215) = 0.99, p = .3$ | | | |
| | $R^2 = 0.10, F(14, 217) = 1,68, p = .06$ | | | $R^2 = 0.13, F(15,216) = 1.41, p = .14$ | | | | | | |

*Note.* M = mediation variable (intolerance of uncertainty), Y = dependent variable (believing), Coeff. = coefficient, SE = standard error, LCI = lower limit of 95% confidence interval, UCI = upper limit of 95% confidence interval, GC = general castes, OBC = other backward castes, SC = scheduled castes, LQ = laterality quotient, X*M = interaction of antecedent and mediation variables (moderation effect); bold values indicate significant indirect effects (not including zero between upper and lower limits of 95% confidence interval).

*** Significant at the .001 level.

** Significant at the .01 level.

* Significant at the .05 level.

The direct effect of the higher caste status (GC vs. others) was similar in believing, behaving, and belonging only (no such effect in bonding). The indirect effect of the lower family income for 30–60 thousand vs. below was similar in believing, bonding, behaving, and belonging. However, the indirect effect of the lower family income for 60 thousand-1 lakh vs. below was similar in believing, bonding, and belonging only (no effect in behaving; although the lower family income had an effect on IU, it failed to translate into an effect on behaving). The direct effect of right-handedness was similar in believing and behaving only (no effect in bonding and belonging). The indirect effect of intermediate level of risk-taking (scoring 3, compared to the lowest level) was similar in believing, bonding, and belonging only (no effect in behaving), whereas the direct effect of intermediate level of risk-taking was in bonding (for score 2, compared to the lowest level) and behaving (for score 4, compared to the lowest level) only (no effect in other dimensions). The direct effect of sex was in behaving only (no effect in believing, bonding, and belonging). Moreover, there was no moderation effect (X-M interaction), except that handedness moderated the effect of IU on believing.

**The interrelationships in the dimensions of religiousness.** The inter-correlations in the dimensions of religiousness (Table 17) for the whole sample varied from .33 to .56. The inter-

**Table 14. The effects (direct and indirect effect) of antecedent and mediation (general IU) variables on bonding in the community sample.**

| Antecedent variable | Effect on M | | | Direct effect on Y | | | Indirect effect on Y | | | |
|---|---|---|---|---|---|---|---|---|---|---|
| | Coeff. | SE | t | Coeff. | SE | t | Coeff. | SE | LCI | UCI |
| Intolerance of uncertainty | | | | 0.08 | 0.04 | 2.19* | | | | |
| Caste (GC vs. others) | 1.35 | 1.34 | 1.00 | -0.97 | 0.76 | -1.28 | 0.11 | 0.13 | -0.11 | 0.40 |
| Caste (OBC vs.SC) | 0.80 | 1.98 | 0.41 | -0.22 | 1.11 | -0.20 | 0.07 | 0.18 | -0.29 | 0.44 |
| X*M (X = caste) | | | | | | | $F(2, 214) = 1.1, p = .3$ | | | |
| Income (above vs. below 1lakh) | 1.30 | 1.93 | 0.67 | 1.68 | 1.08 | 1.55 | 0.11 | 0.18 | -0.26 | 0.50 |
| Income (60 thousand-1Lakh vs. below) | 3.88 | 1.90 | 2.04* | 0.93 | 1.08 | 0.87 | **0.33** | 0.19 | 0.01 | 0.77 |
| Income (30–60 thousand vs. below) | 4.12 | 1.40 | 2.94** | 1.00 | 0.80 | 1.25 | **0.34** | 0.18 | 0.04 | 0.72 |
| Income (10–30 thousand vs. below) | -0.35 | 1.71 | -0.21 | -0.25 | 0.96 | -0.26 | -0.03 | 0.16 | -0.39 | 0.26 |
| X*M (X = income) | | | | | | | $F(4, 212) = 1.1, p = .4$ | | | |
| Handedness (LQ) | 0.001 | 0.014 | 0.036 | 0.014 | 0.008 | 1.75 | 0.000 | 0.001 | -0.002 | 0.002 |
| X*M (X = Handedness) | | | | | | | $F(1, 215) = 0.2, p = .7$ | | | |
| Risk-taking (1vs 2) | -5.23 | 3.31 | -1.58 | -4.21 | 1.87 | -2.25* | -0.44 | 0.34 | -1.28 | 0.03 |
| Risk-taking (1vs 3) | -5.08 | 2.71 | -1.87 | 0.12 | 1.54 | 0.08 | **-0.42** | 0.27 | -1.05 | -0.01 |
| Risk-taking (1vs 4) | -1.22 | 1.83 | -0.66 | 0.12 | 1.03 | 0.12 | -0.10 | 0.17 | -0.48 | 0.21 |
| Risk-taking (1vs 5) | -4.26 | 2.73 | -1.56 | 1.91 | 1.55 | 1.24 | -0.36 | 0.26 | -0.94 | .06 |
| Risk-taking (1vs 6) | -3.82 | 2.18 | -1.75 | 0.50 | 1.23 | 0.40 | -0.32 | 0.25 | -0.86 | 0.11 |
| Risk-taking (1vs 7) | -1.66 | 1.51 | -1.1 | 0.23 | 0.85 | 0.27 | -0.14 | 0.15 | -0.47 | 0.11 |
| X*M (X = Risk-taking) | | | | | | | $F(6, 210) = 0.4, p = .9$ | | | |
| Sex | -0.95 | 1.24 | -0.76 | 1.32 | 0.70 | 1.89 | -0.08 | 0.12 | -0.37 | 0.11 |
| X*M (X = Sex) | | | | | | | $F(1, 215) = 0.11, p = .7$ | | | |
| | $R^2 = 0.10, F(14, 217) = 1.68, p = .06$ | | | $R^2 = 0.12, F(15, 216) = 1.91, p = .02$ | | | | | | |

*Note*. M = mediation variable (intolerance of uncertainty), Y = dependent variable (bonding), Coeff. = coefficient, SE = standard error, LCI = lower limit of 95% confidence interval, UCI = upper limit of 95% confidence interval, GC = general castes, OBC = other backward castes, SC = scheduled castes, LQ = laterality quotient, X*M = interaction of antecedent and mediation variables (moderation effect); bold values indicate significant indirect effects (not including zero between upper and lower limits of 95% confidence interval).

*** Significant at the .001 level.

** Significant at the .01 level.

* Significant at the .05 level.

correlations in the dimensions of religiousness among Hindus (.34 to .57) and Muslims (.06 to .59), among GC (.28 to .60), OBC (.38 to .54), and SC (.11 to .68), and among lower-income (below 30 thousand; .27 to .65), intermediate-income (30 to 60 thousand; .21 to .69), and higher-income (above 60 thousand; .26 to .63) were similar. Moreover, for all these groups, the partial inter-correlations in the dimensions of religiousness (after controlling religiousness) were, mostly, negative and significant. This shows that the dimensions of religiousness were distinct from the general religiousness.

**The relative importance of the dimensions of religiousness in different groups.** We conducted one-way repeated measures analyses of the dimensions of religiousness for the groups (Table 18). Among Hindus, behaving was lower than all other dimensions of religiousness. Among Muslims, there was no difference in the dimensions of religiousness; however, the mean score of believing and behaving tends to be higher than bonding and belonging. Among GC, OBC, and SC, there was no difference in the dimensions of religiousness. Moreover, among the higher-income group (above 60 thousand), there was no difference in the dimensions of religiousness, whereas, among the intermediate-income group (30–60 thousand), believing was higher than bonding and behaving, and, among the lower-income group (below 30 thousand), bonding was higher than all other dimensions of religiousness.

Table 15. The effects (direct and indirect effect) of antecedent and mediation (general IU) variables on behaving in the community sample.

| Antecedent variable | Effect on M | | | Direct effect on Y | | | Indirect effect on Y | | | |
|---|---|---|---|---|---|---|---|---|---|---|
| | Coeff. | SE | t | Coeff. | SE | t | Coeff. | SE | LCI | UCI |
| Intolerance of uncertainty | | | | 0.08 | 0.04 | 1.94 | | | | |
| Caste (GC vs. others) | 1.34 | 1.34 | 1.00 | -1.57 | 0.79 | -1.97* | 0.10 | 0.12 | -0.13 | 0.39 |
| Caste (OBC vs.SC) | 0.80 | 1.98 | 0.41 | -1.72 | 1.17 | -1.47 | 0.06 | 0.17 | -0.29 | 0.45 |
| X*M (X = caste) | | | | | | | $F(2, 214) = 2.73, p = .07$ | | | |
| Income (above vs. below 1lakh) | 1.30 | 1.93 | 0.67 | 1.44 | 1.14 | 1.26 | 0.10 | 0.18 | -0.23 | 0.51 |
| Income (60 thousand-1Lakh vs. below) | 3.88 | 1.90 | 2.04* | 0.53 | 1.13 | 0.47 | 0.30 | 0.21 | -0.01 | 0.77 |
| Income (30–60 thousand vs. below) | 4.12 | 1.40 | 2.94* | 0.88 | 0.84 | 1.05 | **0.32** | 0.20 | 0.001 | 0.78 |
| Income (10–30 thousand vs. below) | -0.35 | 1.71 | -0.21 | -0.45 | 1.01 | -0.45 | -0.03 | 0.15 | -0.36 | 0.28 |
| X*M (X = income) | | | | | | | $F(4, 212) = 1.03, p = .3$ | | | |
| Handedness (LQ) | 0.001 | 0.014 | 0.04 | 0.021 | 0.01 | 2.46* | 0.0 | 0.001 | -0.002 | 0.002 |
| X*M (X = Handedness) | | | | | | | $F(1, 215) = 1.0, p = .3$ | | | |
| Risk-taking (1vs 2) | -5.23 | 3.31 | -1.58 | -3.05 | 1.96 | -1.55 | -0.41 | 0.34 | -1.22 | 0.04 |
| Risk-taking (1vs 3) | -5.08 | 2.71 | -1.87 | -0.08 | 1.61 | -0.05 | -0.40 | 0.27 | -1.03 | 0.02 |
| Risk-taking (1vs 4) | -1.22 | 1.83 | -0.66 | -3.17 | 1.08 | -2.92** | -0.09 | 0.16 | -0.43 | 0.24 |
| Risk-taking (1vs 5) | -4.26 | 2.73 | -1.56 | -0.60 | 1.62 | -0.37 | -0.33 | 0.25 | -0.89 | 0.07 |
| Risk-taking (1vs 6) | -3.82 | 2.18 | -1.75 | 1.17 | 1.30 | 0.91 | -0.30 | 0.25 | -0.83 | 0.13 |
| Risk-taking (1vs 7) | -1.66 | 1.51 | -1.1 | -0.01 | 0.89 | -0.01 | -0.13 | 0.14 | -0.44 | 0.12 |
| X*M (X = Risk-taking) | | | | | | | $F(6, 210) = 0.9, p = .5$ | | | |
| Sex | 0.001 | 0.01 | 0.04 | 2.21 | 0.73 | 3.02** | -0.07 | 0.11 | -0.34 | 0.11 |
| X*M (X = Sex) | | | | | | | $F(1, 215) = 2.46, p = .12$ | | | |
| | $R^2 = 0.10, F(14, 217) = 1.68, p = .06$ | | | $R^2 = 0.18, F(15, 216) = 3.09, p < .001$ | | | | | | |

*Note*. M = mediation variable (intolerance of uncertainty), Y = dependent variable (behaving), Coeff. = coefficient, SE = standard error, LCI = lower limit of 95% confidence interval, UCI = upper limit of 95% confidence interval, GC = general castes, OBC = other backward castes, SC = scheduled castes, LQ = laterality quotient, X*M = interaction of antecedent and mediation variables (moderation effect); bold values indicate significant indirect effects (not including zero between upper and lower limits of 95% confidence interval).

*** Significant at the .001 level.

** Significant at the .01 level.

* Significant at the .05 level.

## Discussion

Consistent with our hypothesis, the present study reports that IU mediates the effect of lower family income on religiousness (in both samples). Earlier studies have suggested that personal uncertainty mediates the relationship between family income and religiousness [6,8]. However, the present study directly tested this likelihood using a personality measure of uncertainty (i.e., IU). Unlike prior studies (reporting mediation effect of psychological defenses in developed countries only; [11]), the present study identifies a psychological variable (i.e., IU) that mediates the relationship of family income with religiousness in a developing country. Moreover, because IU differs between developed and developing countries (discussed in detail by Kumar et al. [35]), it has the potential to explain (mediate) the relationship of lower SES with religiousness across countries, which future studies should focus.

However, the present study also reports that not all effects of the lower family income on religiousness are mediated by IU (i.e., the lower family income is also directly related to religiousness in the students' sample). Recent studies have shown that social class plays an important role in the structuring of self-concept [77], the lower class status is related to lower self-

**Table 16. The effects (direct and indirect effect) of antecedent and mediation (general IU) variables on belonging in the community sample.**

| Antecedent variable | Effect on M | | | Direct effect on Y | | | Indirect effect on Y | | | |
|---|---|---|---|---|---|---|---|---|---|---|
| | Coeff. | SE | t | Coeff. | SE | t | Coeff. | SE | LCI | UCI |
| Intolerance of uncertainty | | | | 0.14 | 0.04 | 3.34** | | | | |
| Caste (GC vs. others) | 1.34 | 1.34 | 1.0 | -2.05 | 0.84 | -2.44* | 0.19 | 0.20 | -0.20 | 0.60 |
| Caste (OBC vs.SC) | 0.80 | 1.98 | 0.41 | -1.96 | 1.24 | -1.58 | 0.11 | 0.30 | -0.50 | 0.74 |
| X*M (X = caste) | | | | | | | $F_{(2, 214)} = 0.14$, $p = .87$ | | | |
| Income (above vs. below 1lakh) | 1.30 | 1.93 | 0.67 | 0.47 | 1.21 | 0.39 | 0.18 | 0.30 | -0.39 | 0.80 |
| Income (60 thousand-1Lakh vs. below) | 3.88 | 1.90 | 2.04* | -0.66 | 1.20 | -0.55 | **0.55** | 0.28 | 0.07 | 1.17 |
| Income (30–60 thousand vs. below) | 4.12 | 1.40 | 2.94** | 0.03 | 0.90 | 0.03 | **0.59** | 0.27 | 0.14 | 1.15 |
| Income (10–30 thousand vs. below) | -0.35 | 1.71 | -0.21 | 0.70 | 1.07 | 0.65 | -0.05 | 0.26 | -0.58 | 0.48 |
| X*M (X = income) | | | | | | | $F_{(4, 212)} = 0.31$, $p = .87$ | | | |
| Handedness (LQ) | 0.001 | 0.014 | 0.036 | -0.002 | 0.009 | -0.226 | 0.000 | 0.002 | -0.003 | 0.004 |
| X*M (X = Handedness) | | | | | | | $F_{(1, 215)} = 0.75$, $p = .4$ | | | |
| Risk-taking (1vs 2) | -5.23 | 3.31 | -1.58 | -2.69 | 2.08 | -1.29 | -0.74 | 0.51 | -1.97 | 0.01 |
| Risk-taking (1vs 3) | -5.08 | 2.71 | -1.87 | 1.14 | 1.71 | 0.67 | **-0.72** | 0.39 | -1.58 | -0.07 |
| Risk-taking (1vs 4) | -1.22 | 1.83 | -0.66 | -1.41 | 1.15 | -1.23 | -0.17 | 0.28 | -0.75 | 0.37 |
| Risk-taking (1vs 5) | -4.26 | 2.73 | -1.60 | 1.10 | 1.72 | 0.64 | -0.61 | 0.38 | -1.42 | 0.07 |
| Risk-taking (1vs 6) | -3.82 | 2.18 | -1.75 | -0.48 | 1.38 | -0.35 | -0.54 | 0.38 | -1.33 | 0.17 |
| Risk-taking (1vs 7) | -1.66 | 1.51 | -1.1 | -0.43 | 0.95 | -0.45 | -0.24 | 0.23 | -0.73 | 0.18 |
| X*M (X = Risk-taking) | | | | | | | $F_{(6, 210)} = 1.28$, $p = .27$ | | | |
| Sex | -0.95 | 1.24 | -0.76 | -0.17 | 0.80 | -0.21 | -0.13 | 0.18 | -0.51 | 0.20 |
| X*M (X = Sex) | | | | | | | $F_{(1, 215)} = 0.53$, $p = .5$ | | | |
| | $R^2 = 0.10$, $F_{(14, 217)} = 1.68$, $p = .06$ | | | $R^2 = 0.09$, $F_{(15, 216)} = 1.41$, $p = .15$ | | | | | | |

*Note.* M = mediation variable (intolerance of uncertainty), Y = dependent variable (belonging), Coeff. = coefficient, SE = standard error, LCI = lower limit of 95% confidence interval, UCI = upper limit of 95% confidence interval, GC = general castes, OBC = other backward castes, SC = scheduled castes, LQ = laterality quotient, X*M = interaction of antecedent and mediation variables (moderation effect); bold values indicate significant indirect effects (not including zero between upper and lower limits of 95% confidence interval).

*** Significant at the .001 level.

** Significant at the .01 level.

* Significant at the .05 level.

esteem [78], and religiousness elevates low self-esteem [79]. Thus, psychological variables like self-esteem may be additional factors mediating the relationship between the lower family income and religiousness. Moreover, because the direct effect of lower family income on religiousness is limited to the students' sample (younger, compared to the community sample), a possibility is that the effect of self-esteem on religiousness is restricted to younger ages.

Similar to the lower family income effect and consistent with the hypothesis, IU mediates the effect of lower caste status on religiousness (in the students' sample). Thus, the effects of lower family income and lower caste status on religiousness may be determined by the same psychological mechanism (i.e., mediated by general IU). However, the present study also reports subtle differences in this common mediator (i.e., in the effect of lower caste status, prospective IU mimics the mediation role of general IU, whereas, in the effect of lower family income, none of the sub-factors of IU replicate the mediation role of general IU). Thus, whereas the active seeking of information for certainty (and not the inhibition of cognition and action) may be the key component in the effect of the lower caste status on religiousness, the unique characteristics of the general IU may be mediating the effect of lower family income (on religiousness). Because the lower family income is related to inhibitive IU (and not the

**Table 17. Simple and partial (in parenthesis; after controlling religiousness) correlations between the dimensions of religiousness in social, economic, and religion groups among the community sample.**

| community | Bel.-Bond. | Bel.-Beh. | Bel.-Blg. | Bond.-Beh. | Bond.-Blg. | Beh.-Blg. |
|---|---|---|---|---|---|---|
| Total (n = 247) | .43 [.32, .55] (-.31 [-.45, -.16]) | .56 [.44, .66] (-.21 [-.36, -.05]) | .39 [.26, .51] (-.34 [-.49, -.20]) | .51 [.40, .62] (-.25 [-.39, -.09]) | .32 [.17, .44] (-.44 [-.56, -.28]) | .41 [.28, .53] (-.43 [-.53, -.31]) |
| Hindu (n = 218) | .43 [.30, .55] (-.33 [-.49, -.17]) | .57 [.46, .66] (-.22 [-.36, -.06]) | .42 [.30, .55] (-.33 [-.48, -.15]) | .50 [.37, .60] (-.20 [-.44, -.15]) | .34 [.19, .47] (-.43 [-.56, -.29]) | .46 [.33, .58] (-.38 [-.51, -.24]) |
| Muslim (n = 20) | .30 [-.18, .75] (-.30 [-.77, .09]) | .34 [-.13, .85] (-.44 [-.78, .19]) | .24 [-.28, .75] (-.27 [-.68, .38]) | .59 [.21, .87] (.13 [-.40, .68]) | .06 [-.45, .49] (-.57 [-.86, -.16]) | .22 [-.41, .62] (-.49 [-.88, .03]) |
| GC (n = 112) | .44 [.27, .60] (-.31 [-.51, -.10]) | .60 [.49, .71] (-.06 [-.23, .11]) | .28 [.08, .48] (-.42 [-.59, -.23]) | .56 [.40, .70] (-.22 [-.44, .01]) | .29 [.08, .49] (-.43 [-.59, -.22]) | .35 [.15, .53] (-.49 [-.64, -.34]) |
| OBC (n = 104) | .49 [.30, .69] (-.22 [-.46, .08]) | .51 [.32, .67] (-.34 [-.56, -.07]) | .38 [.19, .55] (-.45 [-.63, -.20]) | .52 [.34, .69] (-.33 [-.53, -.11]) | .39 [.18, .58] (-.44 [-.61, -.22]) | .54 [.36, .70] (-.22 [-.45, .02]) |
| SC (n = 26) | .11 [-.21, .50] (-.64 [-.85, -.23]) | .54 [.12, .84] (-.17 [-.67, .43]) | .68 [.39, .86] (.14 [-.36, .53]) | .24 [-.10, .63] (-.25 [-.64, .24]) | .24 [-.23, .62] (-.28 [-.69, .18]) | .26 [-.18, .65] (-.74 [-.86, -.58]) |
| Income 1 (n = 50) | .63 [.46, .79] (-.11 [-.41, .22]) | .61 [.37, .78] (-.04 [-.38, .31]) | .33 [.07, .59] (-.53 [-.73, -.29]) | .45 [.20, .67] (-.48 [-.66, -.21]) | .45 [.15, .70] (-.24 [-.52, .11]) | .26 [-.08, .67] (-.53 [-.72, -.28]) |
| Income 2 (n = 68) | .58 [.40, .72] (-.17 [-.45, .14]) | .62 [.48, .76] (-.21 [-.48, .11]) | .39 [.16, .61] (-.32 [-.57, -.03]) | .69 [.53, .82] (-.06 [-.36, .21]) | .21 [-.02, .46] (-.62 [-.75, -.43]) | .34 [.12, .58] (-.50 [-.71, -.21]) |
| Income 3 (n = 127) | .27 [.09, .47] (-.40 [-.59, -.17]) | .65 [.59, .71] (-.17 [-.37, .06]) | .43 [.26, .59] (-.39 [-.58, -.14]) | .37 [.18, .55] (-.36 [-.56, -.17]) | .34 [.14, .51] (-.33 [-.51, -.12]) | .54 [.38, .68] (-.33 [-.48, -.15]) |

*Note*. Bel. = believing, Bond. = bonding, Beh. = behaving, Blg. = belonging, GC = general castes, OBC = other backward castes, SC = scheduled castes, Income 1 = above 60 thousand, Income 2 = 30–60 thousand, Income 3 = below 30 thousand. For a more meaningful understanding (aligned with path analysis), in this analysis, original five categories were merged to form three categories of income. One thousand bootstrap cycles, for estimation of 95% confidence interval, were used to derive correlations. Brackets report 95% confidence interval.

prospective IU; reported in the present study) and the lower SES is related to uncertainty stress [16], perhaps the stress-related component is salient in the general IU, mediating the effect of lower family income on religiousness. Thus, future studies should test the likelihood of such subtle differences between the IU-mediated effects of social (caste) status and family income on religiousness.

In addition, the present study reports a direct relationship between the higher caste status and religiousness (in both samples). Although this perhaps is an initial report, the history and norms of Hindu society support this likelihood (i.e., higher religiousness in higher castes).

**Table 18. The distribution and the comparison of the dimensions of religiousness in different groups among the community sample.**

| | Believing[a] | Bonding[b] | Behaving[c] | Belonging[d] | Comparisons | | | |
|---|---|---|---|---|---|---|---|---|
| | *M (SD)* | *M (SD)* | *M (SD)* | *M (SD)* | *df* | *F* | $\eta^2$ | |
| Hindu | 5.64 (0.11)[c] | 5.73 (0.11)[c] | 5.39 (0.12)[a,b,d] | 5.7 (0.12)[c] | 3, 215 | 3.45* | .05 | b, d, a > c |
| Muslims | 5.53 (0.4) | 5.13 (0.34) | 5.53 (0.44) | 4.58 (0.44) | 3, 17 | 1.5 | .03 | |
| GC | 5.84 (0.14) | 5.76 (0.16) | 5.57 (0.16) | 5.75 (0.17) | 3, 109 | 1.43 | .04 | |
| OBC | 5.57 (0.16) | 5.72 (0.16) | 5.49 (0.17) | 5.57 (0.17) | 3, 101 | 0.62 | .02 | |
| SC | 5.35 (0.32) | 5.63 (0.31) | 4.92 (0.44) | 4.97 (0.41) | 3, 23 | 1.76 | .19 | |
| Income 1 | 5.57 (0.22) | 5.29 (0.25) | 5.12 (0.27) | 5.63 (0.26) | 3, 47 | 2.1 | .12 | |
| Income 2 | 5.89 (0.19)[b,c] | 5.48 (0.23)[a] | 5.33 (0.24)[a] | 5.49 (0.24) | 3, 65 | 3.23* | .13 | a > b, c |
| Income 3 | 5.57 (0.14)[b] | 5.97 (0.12)[a,c,d] | 5.61 (0.14)[b] | 5.61 (0.15)[b] | 3, 124 | 2.68* | .06 | b > d, c, a |

*Note*. Superscripted letters indicate differences between the dimensions of religiousness, significant at the .05 level. GC = general castes, OBC = other backward castes, SC = scheduled castes. Income 1 = above 60 thousand, Income 2 = 30–60 thousand, Income 3 = below 30 thousand. For a more meaningful understanding, in this analysis, original five categories were merged to form three categories of income.

Until the modern era, knowledge of religious texts (of Hindu religion) was limited to higher castes (the scheduled castes were explicitly disqualified; [46]); the religious leaders mostly belong to higher castes; the cultural mobilization (to higher castes' status) entails knowledge of sacred texts or rituals (Sanskratization; [80]), and; higher castes have stronger identification with caste norms [51,73]. Thus, higher castes may have higher religious socialization and involvement, which, in turn, are related to higher religiousness [7,9]. Therefore, religious socialization and involvement may be determining the higher religiousness of the higher castes.

Thus, different SES factors have different patterns of effect on religiousness, some convergent and some divergent (i.e., whereas all the effects of family income on religiousness are negative, the effect of lower caste status is negative and the effect of higher caste status is positive). In response to similar findings (i.e., lower SES individuals have high religiousness and the higher religious engagement attenuates this effect, especially in higher SES individuals), Schieman [9] has suggested that meaning-making is the requirement of all human beings and the poor and rich achieve this through psychological and performative religiousness respectively. However, the present study shows that all the effects of SES (i.e., the positive and the negative effects of caste status, as well as the negative effects of family income) express in all four dimensions of religiousness. Therefore, differences in the nature of religiousness (bonding vs. behaving) may not be an explanation (for the differences of effects in higher caste status vs. lower caste status and lower income status). Alternatively, a simple explanation may be that religiousness is latent (i.e., residing in the basic human nature or human psychology), and different factors only differently initiate its expression. The cognitive theory of religiousness supports this explanation (i.e., religiousness is based on the basic human cognitive abilities [81]). Moreover, findings on the genetic predisposition for religiousness [82,83] and the brain localization of religiousness [64,65] also support this likelihood. Thus, any of the factors, like the higher level of socialization and participation or the higher level of IU (determined by the lower social status or by the lower income), may initiate a similar form of religiousness (i.e., expressing similarly in all four dimensions of religiousness) independently.

In addition, the above-discussed findings are important for understanding the Indian socio-cultural system. The directionally opposite effects (suppression effect [84]) of caste status on religiousness (and its dimensions) support a nuanced model of the relationship of the hierarchical caste system with religiousness. Forced by the unpleasantness of IU (i.e., fear of the unknown), the approach of the lower castes towards religion is likely to be similar to that of solution seekers, whereas, determined by the socialization and participation, the approach of the upper castes towards religion is likely to be similar to that of professionals (i.e., leaders, helpers, guide, organizers, or thinkers). Therefore, there may be a positive religious relationship between the lower and higher castes (i.e., client-provider relationship). Moreover, endorsement of the caste system by the religion (by the Hindu scriptures; [46]), may establish a mutually reinforcing relationship between religiousness and caste hierarchy. Thus, because the positive religious relationship adds an approach-tendency to the approach-avoidance conflict of the lower castes towards the upper castes [85,86], this model explains the paradoxical behavior of the lower castes (i.e., behavior of not leaving the fold of Hindu religion, despite its support for the discriminatory caste system [85]). Moreover, because the caste characteristics, similar to India, have been reported in other countries [85], and are evident in cross-national affairs also [87], the present findings on the religiousness of caste may be of a wider cross-national significance.

Studies have documented higher little traditions (i.e., beliefs and rituals with local and individualistic outlook) in lower castes and higher greater traditions (i.e., beliefs and rituals with orthodox and universal outlook) in upper castes [88]. Thus, some differences in the

religiousness of the lower and higher castes may be suspected. The above-discussed client-provider role differences (between lower and higher castes) also support this likelihood. However, the present study shows that the form of religiousness (i.e., expression in all four dimensions) is similar in the effects of lower (IU-mediated) and higher (socialization-based) caste statuses (discussed above) and there is no difference in the salience of the dimensions of religiousness among castes. Thus, the little and greater traditions (or client-provider role differences) may not affect the nature and quality of religiousness. However, as far as we know, the studies on the greater and little traditions have not controlled the effect of income, which, otherwise, is different between the lower and upper castes [47]. Moreover, the present study shows that the quality of religiousness is different between the higher and lower income groups. Thus, perhaps family income is a factor in any likely qualitative difference between the religiousness of the greater and little tradition communities. Future studies should test these likelihoods.

We failed to support the hypothesized relationship between education and religiousness in the present study. Perhaps because earlier studies have reported a weak (negative) relationship between education and religiousness [7]; there is a narrower band of education levels in the students' sample (larger sample), and; the sample size is smaller in the community sample, the present study has failed to capture the effect of education on religiousness.

Consistent with prior studies [15], we found that the majority-religion community (Hindus) was less religious than Muslims (demographically second-largest religion; except in bonding) and other minorities (including Jain, Buddh, Sikh, etc; only among students). However, unlike prior studies [15], the present study fails to support social exclusion as a factor in the higher religiousness of minorities (i.e., because there is a lack of IU-mediation in this relationship). Moreover, because the pattern of the salience of dimensions of religiousness is different between Hindus (bonding is salient) and Muslims (no difference in the dimensions of religiousness), perhaps, along with the effect on degrees, religion also affect the quality of religiousness. However, unlike prior studies [45], the present study fails to find a salience of behaving dimension, as well as a strong positive inter-relationship in the dimensions of religiousness, among Muslims (following a monotheistic religion). Instead, Muslims have weaker interrelationships and negative partial inter-correlations (after controlling general religiousness) in the dimensions of religiousness, similar to Hindus (a polytheistic eastern religion) or Indians, in general (which we hypothesized to have lower inter-correlations). Therefore, instead of religion, the weak, or negative, interrelationship in the dimensions of religiousness seems to be the characteristics of the socio-cultural system (of India).

Along with the effect of socio-cultural factors, the present study reports a complex pattern of the effects of bio-socio-developmental factors on religiousness. For example, in the students' sample, IU mediates the effect of age, handedness, height (in women only), and risk-taking on religiousness, whereas in the community sample, except for risk-taking, there is no such IU-mediated effect. Perhaps the smaller sample size may be a factor in the null-effect report for the community sample. However, alternatively, age-related developmental changes may be the determinant. In the present study, the community sample is older than the students' sample; age is related to an increase in IU among adolescents' (but, perhaps, not among adults [56]; similar age effects occur for ambiguity aversion also [89]), and; religiousness increases with age up to young adulthood [55], and the relationship thereafter flattens (no relationship; [90]). Thus, the effects of age, height, and handedness may be affected by the age-related variations of IU and religiousness. However, because, consistent with prior studies [60], we found a direct effect of right-handedness on religiousness in the community sample, the effect of the age-related variations, on the handedness (effect), may be slightly different.

The pattern of effects of risk-taking on religiousness is quite interesting, i.e., at the intermediate levels of self-reported risk-taking, the effects on religiousness are negative (the IU-

mediated indirect effect in the students' sample, and the direct as well as IU-mediated indirect effects in the community sample), whereas, at the highest level of self-reported risk-taking, the effect on religiousness is positive (direct effect in the students' sample). Earlier reports have supported both positive [59] and negative [91,92] effects of self-reported risk-taking on religiousness. Thus, because self-reported risk-taking is a genetically determined trait [76], future studies should focus on delineating the different patterns of the effect of the different levels of risk-taking on religiousness i.e., the IU-mediated vs. direct effects and the positive vs. negative effects of different levels of self-reported risk-taking on religiousness.

Moreover, sex has no IU-mediated effect, whereas, consistent with the prior reports [57], there is a direct effect of sex on religiousness. Because this effect similarly occurs in both samples, perhaps the age-related developmental variations are not involved in the effect of sex on religiousness. However, because the effect of sex is consistent only in the behaving dimension of religiousness across samples, some subtle age-related differences may be suspected. Future studies should explore these likelihoods. In addition, we failed to find the relationship between BMI and religiousness. Earlier studies have reported an inconsistent pattern of relationship between BMI and religiousness [62].

Noticeably, for the above-mentioned bio-socio-developmental factors, the present study reports different patterns of the effect on religiousness, i.e., direct effects, indirect effects (mediated by IU), positive effects, and negative effects. Moreover, the expression of each effect is largely similar across the four dimensions of religiousness (especially in the larger size students' sample). Thus, the findings for the bio-socio-developmental factors, in the present study, additionally support the contention that religiousness is a latent variable that varied factors may independently initiate.

In the present study, the analyses of the sub-factors of IU show that the different antecedent variables have varied effects on the sub-factors of IU. For example, family income has an effect on inhibitive IU only, caste status has an effect on prospective IU only, and age has effects on both prospective and inhibitive IUs (in the students' sample). Thus, it is difficult to explain the determinants of such effects of different socioeconomic and biosocial factors on different sub-factors of IU. Perhaps, there may be some unknown sub-components in both the socioeconomic and the bio-socio-developmental factors that are related to either prospective IU or inhibitive IU, which future studies should explore.

Moreover, the present study shows that only prospective IU mediates the effect of antecedent variables on religiousness. Earlier studies have shown that prospective IU (and not inhibitive IU) is related to psychological relief-based motives of religiousness [38]. Thus, the psychological relief motive in the active seeking of information for certainty (i.e., prospective IU) may be a determinant of religiousness. Furthermore, because inhibitive IU is not related to religiousness, the inhibitive response in the face of uncertainty may not be a factor in religiousness (except, perhaps in the effect of family income on religiousness, discussed above). However, inhibitive IU is related to depression [93] and studies have documented an inconsistent pattern of relationship between religiousness and depression [94]. Thus, based on the present study, we suggest that, by controlling the effects of prospective and inhibitive IUs, inconsistency in the reported patterns of relationship between religiousness and depression may be resolved.

In the present study, we have not elaborated, further, the few odd findings of moderation effects (i.e., the moderation effect of risk-taking on the relationship of IU with believing, in the students' sample, and the moderation effect of handedness on the relationship of IU with believing, in the community sample), because it may be beyond the scope of the present study. Moreover, although, we considered IU as an antecedent of religiousness the opposite

relationship is also a possibility. However, again, studying this is beyond the scope of the present study.

The present study may have several limitations. First, instead of actual assessment, the income assessment is self-reported. There may be differences in the actual and self-reported incomes. However, the single item query and the response in terms of the choice of income brackets are suggested to be good procedures for income surveys [95]. Moreover, the demography of the reported income brackets is similar to that reported by the national sample surveys of India [96]. Second, the height and weight are self-reported and not measured. However, because studies have shown that self-reported height and weights correspond with actual height and weights [97], this is not likely to be a significant confounding factor. Third, whereas religiousness may differ along the rural-urban dimension, we have not gathered information about urban-rural dwellings and the nature of family (nuclear vs. joint). However, because the present study was conducted in an urban setting we expect that urban dwelling and nuclear family may be the norm. Lastly, because both IU and religiousness are measured through questionnaires, the assignment of these variables as antecedents and consequents is theory-driven and not supported by methodological procedures (not an experiment).

Thus, in conclusion, the present study comprehensively studies IU-mediated effects (and direct effects also) of a large number of antecedent (environmental and bio-socio-developmental) variables on religiousness and its dimensions. It shows that IU mediates the effects of lower family income and lower caste status on religiousness; the higher caste status is related to higher religiousness; bio-socio-developmental factors have varied effects (direct, IU-mediated, positive, and negative) on religiousness, and; the most of the above-mentioned effects have similar expression in all four dimensions of religiousness. Thus, the present study supports the contention that religiousness is a latent variable that varied factors may independently initiate. It also suggests a nuanced model of the relationship between caste hierarchy and religiousness and reports qualitative differences in religiousness between Hindus vs. Muslims and between lower vs. higher income groups. The similar low inter-correlations (or negative partial inter-correlations) between the religiousness dimensions of the monotheistic (Muslims) and polytheistic (Hindus) Indians is another significant finding of the present study. Moreover, by using a standardized personality measure of personal uncertainty (i.e., IU) and a standardized, cross-culturally valid, and multidimensional measure of religiousness, the present study presents a new, and richer, approach to the study of the relationship between SES and religiousness.

## Supporting information

**S1 Fig. The conceptual path diagram of the studied relationships of height (other variables as covariates) with IU and religiousness separately for men and women in the students' sample.** IU = intolerance of uncertainty, GC = general castes, OBC = other backward castes, SC = scheduled castes, th. = thousand, LQ = laterality quotient (handedness). (DOCX)

**S2 Fig. The conceptual path diagram of the studied relationships of antecedents with IUP, IUI, and religiousness in the students' sample.** IUP = prospective intolerance of uncertainty, IUI = inhibitive intolerance of uncertainty, GC = general castes, OBC = other backward castes, SC = scheduled castes, th. = thousand, LQ = laterality quotient (handedness). (DOCX)

**S3 Fig. The conceptual path diagram of the studied relationships of antecedents with IUP, IUI, and religiousness in the community sample.** IUP = prospective intolerance of

uncertainty, IUI = inhibitive intolerance of uncertainty, GC = general castes, OBC = other backward castes, SC = scheduled castes, th. = thousand, LQ = laterality quotient (handedness). (DOCX)

**S1 File.**
(XLSX)

**S2 File.**
(XLSX)

## Acknowledgments

The authors thank Professor Heinz Streib for comments on an earlier version of this manuscript and acknowledge the help rendered by Dr. Reena Saini and Dr. Ranjeeta Jain in data collection.

## Author Contributions

**Conceptualization:** Sanjay Kumar.

**Formal analysis:** Sanjay Kumar, Martin Voracek.

**Methodology:** Martin Voracek.

**Writing – original draft:** Sanjay Kumar.

**Writing – review & editing:** Sanjay Kumar, Martin Voracek.

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
