## [Decision Letter · Decision Letter 0]

20 May 2022

PONE-D-21-30575The relationships of family income and caste-status with religiousness: Mediation effect of intolerance of uncertaintyPLOS ONE

Dear Dr. Kumar,

Thank you for submitting your manuscript to PLOS ONE. After careful consideration, we feel that it has merit but does not fully meet PLOS ONE’s publication criteria as it currently stands. Therefore, we invite you to submit a revised version of the manuscript that addresses the points raised during the review process.The research question needs to be clarified. Without a clearly articulated research question, this paper is very difficult to read. The authors indicate that this study examines the effect of socioeconomic status (SES) on intolerance of uncertainty (IU; a personality measure of self uncertainty) and religiousness. This suggests that IU and religiousness are the dependent variables, and SES is the independent variable. However, the authors mentioned the mediating effect of IU on the association between lower family income and lower caste status. What is the focus of the paper? You seem to switch your focus to the association of SES and religiousness mediated by the IU. If that’s your focus, you should reposition the paper.

It is not clear why the association of SES, IU and religiousness is important. You need to provide a stronger motivation about your research question, and provide some theoretical background about how these variables affect each other. Instead of measuring different variables and report the results in different tables, you should think about building a model and map out the associations/effects. This will help the readers understand what you try to do in the paper.

On page 26, you mentioned that “developmental-genetic processes related to age, height, and hand preference may affect brain structures of IU and religiousness to determine the interrelationships”. You measured these variables, but did not provide a clear explanation how they affect IU and religious, and why they are important and relevant to your study. If IU and religious are genetically determined, then why do you investigate SES at the first place? How does that relate to the initial research question, which seems to investigate the effect of SES?  

You need to clearly define religiousness. Could you provide a review of prior literature on this concept? How is defined? How is it measured? Is an attitude or a personal trait? You mentioned its dimension include believing, bonding, behaving, and belonging. Could you elaborate on these factors? Without a clear lit review, it is difficult to see how your study extend prior work and determine its contribution.

You need provide explanation about the caste status. Don’t assume readers know it. How is this system determined? How is tracked? Is it still relevant today? If relevant, what is its impact? Is it correlated with family income? It appears to be a self-reported variable in your two samples. How do you make sure they reported this variable truthfully?

You should provide a clear interpretation of your results. For example, you find the positive effect of self-reported risk-taking (in highest risk-takers) on religiousness in one sample but the negative effect in the different sample. Could you explain why this happens in your samples and provide a reconciliation of the difference? 

We look forward to receiving your revised manuscript.

Kind regards,

Ning Du

Academic Editor

PLOS ONE

**Journal requirements:**

Reviewers' comments:

Reviewer's Responses to Questions

**Comments to the Author**

1. Is the manuscript technically sound, and do the data support the conclusions?

Reviewer #1: No

2. Has the statistical analysis been performed appropriately and rigorously? 

Reviewer #1: Yes

3. Have the authors made all data underlying the findings in their manuscript fully available?

Reviewer #1: No

4. Is the manuscript presented in an intelligible fashion and written in standard English?

Reviewer #1: Yes

5. Review Comments to the Author

Reviewer #1: This study was actually conducted in the spirit of a theoretical lineage left untapped by the authors, namely the opiate thesis of religion. This negligence of the literature shows that the authors have relied on ramifications of a key theory instead of its main source. Consequently, the authors have measured their focal concepts, ses and religiousness, in a manner at odd with the state of the art literature on religion. The inclusion of handedness as some sort of correlated factor of religiousness is ludicrously void of any theoretical support and may likely come from the p-hacking of the small scale psychological studies being cited. Measuring socioeconomic status as income alone also gives away how much the authors are unfamiliar with the prevailing evidence on the multiaxial nature of ses and class. Finally, religion literature has already made popular the knowledge that religiosity comprises of beliefs, behaviors, and identification, the remaking of the wheel here still adheres to the proclamation of religiousness as a focal research question, instead of all other aspects of religiosity. The sample is composed of two very different sources in the population, whose unequal representativeness cannot even be redeemed by sample weights, let alone the absence of it.

6. PLOS authors have the option to publish the peer review history of their article (what does this mean?). If published, this will include your full peer review and any attached files.

Reviewer #1: No

---

## [Author Response · Author response to Decision Letter 0]

28 Jun 2022

The Editor: Thank you for submitting your manuscript to PLOS ONE. After careful consideration, we feel that it has merit

The Authors: Thank you for your evaluation and for your gentle guidance to conduct a wholesome discussion of our results.

The Editor: The research question needs to be clarified. Without a clearly articulated research question, this paper is very difficult to read. The authors indicate that this study examines the effect of socioeconomic status (SES) on intolerance of uncertainty (IU; a personality measure of self uncertainty) and religiousness. This suggests that IU and religiousness are the dependent variables, and SES is the independent variable. However, the authors mentioned the mediating effect of IU on the association between lower family income and lower caste status. What is the focus of the paper? You seem to switch your focus to the association of SES and religiousness mediated by the IU. If that’s your focus, you should reposition the paper.

The Authors: Thank you for pointing this out. We have modified the manuscript accordingly to emphasize the mediation role of IU. Please see the introduction, results, and discussion sections. 

The Editor: It is not clear why the association of SES, IU and religiousness is important. You need to provide a stronger motivation about your research question, and provide some theoretical background about how these variables affect each other.

The Authors: We have tried to formulate our justification for studying the mediation role of IU in the effect of SES on religiousness. Along with reporting inadequacies in the earlier suggested mediation factors, we have also provided a theoretical connection for the interrelationships of SES, IU, and religiousness. Please see the introduction section 

The Editor: Instead of measuring different variables and report the results in different tables, you should think about building a model and map out the associations/effects. This will help the readers understand what you try to do in the paper.

The Authors: Thank you for the suggestion. We have tried to organize our discussion. Please see the discussion section

The Editor: On page 26, you mentioned that “developmental-genetic processes related to age, height, and hand preference may affect brain structures of IU and religiousness to determine the interrelationships”. You measured these variables, but did not provide a clear explanation how they affect IU and religious, and why they are important and relevant to your study. 

The Authors: Thank you for pointing this out. We have now added this all to the introduction section. We have reviewed relationships of different bio-socio-developmental factors with both IU and religiousness. Moreover, because studies support the mediation role of brain structure in determining the relationship of bio-socio-developmental factors (and the environmental factors also) with IU-religiousness connection, perhaps the mediation role of brain structure justifies the study of diverse determinants of the IU-religiousness connection. Please see lines 215-234.

The Editor: If IU and religious are genetically determined, then why do you investigate SES at the first place? How does that relate to the initial research question, which seems to investigate the effect of SES? 

The Authors: There seems to be some misunderstanding. We only contended that both IU and religiousness are related to the prefrontal cortex. In this revised draft we have tried to be more specific and unambiguous in our language and have laid the justification for the relationship of a wide variety of antecedent variables (the bio-socio-developmental and the environmental variables) with the IU-religiousness connection. Please see lines 222-234

The Editor: You need to clearly define religiousness. Could you provide a review of prior literature on this concept? How is defined? How is it measured? Is an attitude or a personal trait? You mentioned its dimension include believing, bonding, behaving, and belonging. Could you elaborate on these factors? Without a clear lit review, it is difficult to see how your study extend prior work and determine its contribution.

The Authors: Thank you for the suggestion. We have addressed these questions in this revision. Please see the introduction section, lines 56-62, 161-183.

The Editor: You need provide explanation about the caste status. Don’t assume readers know it. How is this system determined? How is tracked? Is it still relevant today? If relevant, what is its impact? Is it correlated with family income? It appears to be a self-reported variable in your two samples. How do you make sure they reported this variable truthfully?

The Authors: Please see the introduction section, lines 186-205. The relevant references deal with these questions in detail. However, we are also answering them here, for clarity.

1. Caste is decided by birth. Thus, the caste of a person is the caste of his parents, especially of his father.

2. Caste is an endogamous group fitted to a hierarchy in the broader society. Thus birth to a specific group decides the ritual and social status of an individual. Moreover, because hierarchy is also strengthened by the access to resources (occupation-caste connection), economic asymmetry is there to support caste hierarchy.

3. Yes, it is relevant and important. Despite some changes in the caste-occupation connection, caste is a psycho-social reality of modern India. Endogamy is still a predominant theme in marriages (less than 6% are inter-caste marriages, https://en.wikipedia.org/wiki/Inter-caste_marriages_in_India), caste consciousness is there, and even caste hierarchy also finds an explicit expression occasionally (https://www.hindustantimes.com/india-news/crimes-against-dalits-tribals-increased-in-covid-pandemic-year-ncrb-101631731260293.html). We have added new references to support this. Please see lines 193-196.

4. Yes, irrespective of modern economic opportunities, wealth differences occur across caste statuses and it is well documented. E.g., Zacharias & Vakulabharanam, 2011 

5. Caste is a reality of Indian life. Generally, in a particular small society, everyone has an idea of each other’s caste. Thus, it is quite difficult to fake the caste. Moreover, because caste is a criterion for deciding the disbursement of benefits of various government schemes, there is an incentive to exactly report caste category (GC, OBC, SC), even for lower caste individuals. Asking participants about their caste category is an established common procedure for determining the caste categories, followed by earlier studies also (e.g., Sankaran et al., 2017; Zacharias & Vakulabharanam, 2011). Please see lines 271-275.

The Editor: You should provide a clear interpretation of your results. 

The Authors: Thank you for the suggestion. We have tried improvement. Please see the discussion section.

The Editor: For example, you find the positive effect of self-reported risk-taking (in highest risk-takers) on religiousness in one sample but the negative effect in the different sample. Could you explain why this happens in your samples and provide a reconciliation of the difference? 

The Authors: There seems to be some misunderstanding. To clarify, we found a negative relationship between risk-taking and religiousness for the intermediate level of risk-taking, in both samples (students and community), and a positive relationship between risk-taking and religiousness for the highest level of risk-taking, in the students' sample only. Thus, the negative relationship may be associated with the intermediate level of risk-taking and the positive relationship with the highest level of risk-taking. We have tried to be more specific in reporting these results. Please see the discussion section.

The Reviewer: This study was actually conducted in the spirit of a theoretical lineage left untapped by the authors, namely the opiate thesis of religion. This negligence of the literature shows that the authors have relied on ramifications of a key theory instead of its main source. 

The Authors: Thank you for this suggestion. In this revised manuscript, we have reviewed psychological studies that support the contention that the constraints and inadequacies of low SES lead to religiousness. We have discussed the strong (i.e., constraints and inadequacies generate religiousness) as well as the weak (i.e., constraints and inadequacies increase religiousness) versions of this contention. Please see the introduction section, lines 60-72.

The Reviewer: Consequently, the authors have measured their focal concepts, ses and religiousness, in a manner at odd with the state of the art literature on religion.

The Authors: The reviewer has not clarified how he got such an impression. We have not framed a composite SES variable (it has conceptualization difficulties). Instead, we simultaneously entered the different indices of SES, i.e., family income, caste status, and education level, and assessed their effect on religiousness. Income, social status, and education are commonly used indexes of SES. Moreover, we have used standardized measures of religiousness and IU. 

The Reviewer: The inclusion of handedness as some sort of correlated factor of religiousness is ludicrously void of any theoretical support and may likely come from the p-hacking of the small scale psychological studies being cited. 

The Authors: In response to this question related to the lack of support for the study of the effect of handedness and other bio-socio-developmental variables on religiousness, we have attempted a justification. Please see our response to the editor. Please see the lines 215-234 also.

The Reviewer: Measuring socioeconomic status as income alone also gives away how much the authors are unfamiliar with the prevailing evidence on the multiaxial nature of ses and class. 

The Authors: Perhaps the reviewer has not noticed that caste and education (both indices of SES) were also studied. We have rewritten the relevant text to highlight this, please see the lines 205-212, 238-241.

The Reviewer: Finally, religion literature has already made popular the knowledge that religiosity comprises of beliefs, behaviors, and identification, the remaking of the wheel here still adheres to the proclamation of religiousness as a focal research question, instead of all other aspects of religiosity.

The Authors: Thank you for the suggestion. 

In the present study, we analyzed religiousness in terms of four basic dimensions of believing, bonding, behaving, and belonging. We also analyzed the quality of religiousness by comparing the salience of the dimensions. 

However, now, we have derived additional conclusions based on these analyses. Please see the lines 174-184, 592-605.

The Reviewer: The sample is composed of two very different sources in the population, whose unequal representativeness cannot even be redeemed by sample weights, let alone the absence of it.

The Authors: Please see that the study was conducted in the discovery-replication sample approach, in which replication needs to be done in a different population. Therefore, sample weighting may be irrelevant.

---

## [Editor Report · Decision Letter 1]

11 Jul 2022

PONE-D-21-30575R1The relationships of family income and caste-status with religiousness: Mediation role of intolerance of uncertaintyPLOS ONE

Dear Dr. Kumar,

Thank you for submitting your manuscript to PLOS ONE. After careful consideration, we feel that it has merit but does not fully meet PLOS ONE’s publication criteria as it currently stands. Therefore, we invite you to submit a revised version of the manuscript that addresses the points raised during the review process.

I only see section titles for Abstract, Method and Discussion. Please include tittles for other sections.  Please ensure that your decision is justified on PLOS ONE’s publication criteria and not, for example, on novelty or perceived impact.

We look forward to receiving your revised manuscript.

Kind regards,

Ning Du

Academic Editor

PLOS ONE

Journal Requirements:

Additional Editor Comments:

I only see sections titles for Abstract, Method and Discussion. Please include section titles for other parts.
---

## [Author Response · Author response to Decision Letter 1]

15 Jul 2022

The Editor: I only see section titles for Abstract, Method and Discussion. Please include tittles for other sections.

The Authors: Done. Please see the marked changes in the manuscript.

---

## [Editor Report · Decision Letter 2]

4 Aug 2022

The relationships of family income and caste-status with religiousness: Mediation role of intolerance of uncertainty

PONE-D-21-30575R2

Dear Dr. Kumar,

We’re pleased to inform you that your manuscript has been judged scientifically suitable for publication and will be formally accepted for publication once it meets all outstanding technical requirements.

Kind regards,

Ning Du

Academic Editor

PLOS ONE
---

## [Editor Report · Acceptance letter]

17 Aug 2022

PONE-D-21-30575R2 

The relationships of family income and caste-status with religiousness: Mediation role of intolerance of uncertainty   

Dear Dr. Kumar:

I'm pleased to inform you that your manuscript has been deemed suitable for publication in PLOS ONE. Congratulations! Your manuscript is now with our production department. 

Kind regards, 

on behalf of

Dr. Ning Du 

Academic Editor

PLOS ONE